# How to Train a Leader: Hierarchical Reasoning in Multi-Agent LLMs

## Abstract

Large Language Models (LLMs) have achieved strong performance on a wide range of complex reasoning tasks, yet further gains are often possible by leveraging the complementary strengths of multiple models. While multi-agent frameworks can improve solution quality by leveraging multiple LLMs, existing methods are often computationally expensive, both at training and inference time. In this work, we introduce a hierarchical multi-agent framework that addresses these challenges by training only a single leader LLM to coordinate a team of untrained peer agents. To this end, we propose **M**ulti-agent guided **L**eader **P**olicy **O**ptimization (MLPO), a novel approach which trains the leader to evaluate and synthesize agent responses without auxiliary value networks or explicit agent feedback. Leaders trained with MLPO exhibit improved performance not only when interacting with the agent team at inference time, but also enjoy improved performance when deployed in single-agent settings without the team. Empirical results on BBH, MATH, and MMLU demonstrate that our framework achieves substantial performance improvements over both single-agent and multi-agent baselines. Our results highlight the effectiveness and efficiency of training a single, flexible leader for collaborative reasoning in multi-agent LLM systems.

## 1 Introduction

Large Language Models (LLMs) have rapidly become foundational in natural language processing, demonstrating remarkable versatility across tasks ranging from translation to complex question answering (Brown et al., 2020; Bommasani et al., 2021). While these models have shown impressive performance in a wide range of tasks, they still struggle with both factual accuracy and complex reasoning (Huang et al., 2023; Mondorf & Plank, 2024).

To address some of these limitations, multi-agent approaches in particular have shown promise: by having two or more LLMs collaboratively solve a problem, the group of models may collectively produce better answers than a single model alone (Du et al., 2023; Chen et al., 2023; Wu et al., 2023; Eo et al., 2025; Motwani et al., 2025; Jin et al., 2025; Estornell et al., 2024b; Li et al., 2024b; Liu et al., 2024a; Khan et al., 2024; Estornell & Liu, 2024; Chang, 2024; Feng et al., 2025b;a; 2024a; Chen et al., 2025; Wang et al., 2024; Li et al., 2024a; Feng et al., 2024b; Chan et al., 2023). However, current multi-LLM frameworks mostly rely on off-the-shelf models that have not been explicitly trained to collaborate, treating effective collaboration as an emergent property of large models (Du et al., 2023; Chen et al., 2023; Yang et al., 2025a; Khan et al., 2024). These works assume that general-purpose LLMs are inherently capable of debating, verifying, or correcting each other, and focus on designing (or training) mechanisms to better elicit this behavior from off-the-shelf models.

In contrast, some recent work has begun to explore training schemes that explicitly tune models for multi-agent collaboration (Estornell et al., 2024a; Zhou et al., 2025; Li et al., 2025a; Wan et al., 2025; Motwani et al., 2025; Qiu et al., 2024; Park et al., 2025). For example, ACC-Collab (Estornell et al., 2024a) jointly trains a two-model team (one actor and one critic) to solve tasks through iterative dialogue. Other works have extended this paradigm to teams of two or three similarly specialised models (Zhou et al., 2025; Li et al., 2025a; Wan et al., 2025; Motwani et al., 2025), demonstrating that training specialised teams can yield performance improvements. Some approaches further scale to larger groups of trained agents (Qiu et al., 2024; Park et al., 2025). However, a key drawback of

each of these methods is that they require jointly optimising multiple LLMs, an approach that is computationally expensive and difficult to scale. This raises a fundamental question:

*Can we enhance a multi-agent team's collaborative reasoning by training **only a single** model, rather than training multiple models?*

In this paper, we answer this question by proposing a novel hierarchical multi-agent architecture for collaborative reasoning, **M**ulti-agent guided **L**eader **P**olicy **O**ptimization (MLPO). Our approach draws inspiration from hierarchical structures in multi-agent systems, where a designated leader coordinates a team of subordinate (Hong et al., 2024; Liu et al., 2024b; Singh et al., 2024; Jin et al., 2025; Li et al., 2025b), and from recent advancements in reinforcement-learning-based reasoning techniques (Zhu et al., 2024). Specifically, we introduce a framework in which a single *leader* LLM is trained to solve tasks with the assistance of a team of untrained (off-the-shelf) LLM agents that provide candidate solutions. During inference, the leader queries the agent team, aggregates their outputs, and synthesises a final answer.

MLPO can be regarded as multi-agent *guided* training. During training time, the leader is exposed to multiple candidate solutions provided by the agent team, which exposes the leader to a richer pool of candidate solutions. This additional exposure gives the leader opportunities to broaden its search space during the RL phase of MLPO.

Since our approach involves training only a single model, our multi-agent guided training scheme offers clear advantages, including improved efficiency during training and greater flexibility at inference time. MLPO has several key advantages. Firstly, only a single model needs to be trained, greatly reducing training cost compared to methods which train multiple models (Estornell et al., 2024a; Motwani et al., 2025; Park et al., 2025). Secondly, unlike previous methods, which require all agents to participate at test time, we discover that our trained leader can also function independently.

We empirically observe that our training pipeline not only enhances the leader's performance when interacting with the agent team at inference time, but also enhances its performance as an individual model operating without the team. In other words, our training approach enhances both the leader's collaborative multi-agent performance and its zero-shot performance (i.e., as a single model without the agent team). Consequently, when inference-time cost is a constraint, the leader alone can still achieve competitive results, although full collaboration provides the best performance.

We evaluate our hierarchical leader team architecture using *7–9B*-parameter models on three reasoning benchmarks: BIG-BENCH HARD (BBH) (Suzgun et al., 2022), MATH (Hendrycks et al., 2021), and MMLU (Hendrycks et al., 2020). We consider two classes of baselines (i) strong *prompt-only* methods such as CoT or multi-agent debate with off-the-shelf LLMs (Wei et al., 2022; Du et al., 2023), and (ii) *trained* variants that employ the same agents but use different training techniques and inference (Estornell et al., 2024a; Maurya et al., 2024; Kumar et al., 2024; Zhu et al., 2024).

The contributions of this paper are as follows:

- **Hierarchical framework**: We propose a hierarchical multi-agent architecture where a single leader LLM is trained to coordinate a team of untrained LLMs for collaborative reasoning.
- **Multi-agent guided GRPO objective**: We introduce a GRPO-based approach, coined MLPO, enabling the leader model to effectively refine solutions proposed by the agent team.
- **Strong empirical results**: Extensive experiments demonstrate that our hierarchical multi-agent approach significantly outperforms existing baselines, including both trained and untrained single-agent and multi-agent methods.
- **Systematic ablation study**: We perform thorough ablations on team composition, aggregation methods, and sampling strategies, providing insights into their individual contributions and guiding further improvements to our framework.

## 2 RELATED WORK

**Multi-LLM Collaboration**   Similar to our proposed method, numerous works have proposed the use of multiple LLMs to improve answer quality and accuracy. These works can be loosely grouped into two categories: *collaborative methods*, which engage multiple LLMs in collaboration, often in

the form of iterative discussion (Du et al., 2023; Estornell et al., 2024b; Li et al., 2024b; Liu et al., 2024a; Liang et al., 2023; Khan et al., 2024; Estornell & Liu, 2024; Chang, 2024; Feng et al., 2025b;a; 2024a; Li et al., 2023; Ye et al., 2025; Yang et al., 2025b), and *aggregation methods* which attempt to bootstrap multiple LLMs (responses) together without direct collaboration (Dai et al., 2024; Yang et al., 2025a; Jiang et al., 2023; Huang et al., 2024). There are some methods that fall into both categories, such as mixture of agents (MoA) (Chen et al., 2025; Wang et al., 2024; Li et al., 2024a).

Several unique paradigms have been proposed within collaborative multi-LLM approaches. Notably, Du et al. (2023) proposes a system of multi-agent debate in which a team of LLMs directly communicates with one another over rounds of discussion. Other works have extended this framework with a focus on designing mechanisms to enhance the collaborative ability of off-the-shelf-models (Khan et al., 2024; Chen et al., 2023; Zhang et al., 2024; Li et al., 2023; Liang et al., 2023; Feng et al., 2025b;a; 2024a; Li et al., 2023; Ye et al., 2025; Yang et al., 2025b; Liu et al., 2024b). While effective in certain settings, these approaches do not directly train models to collaborate instead aiming to improve the collaboration of off-the-shelf models. Diverging from this work, our proposal involves training a reasoning-based leader LLM which explicitly guides the team towards the right direction.

More recently, several recent works have proposed training schemes to directly enhance the collaborative ability of LLMs. In particular, ACC-Collab (Estornell et al., 2024a) designs a scheme for training a two model team consisting of an actor and critic agent. Building upon this paradigm, other works have also designed two-agent (Zhou et al., 2025; Li et al., 2025a; Wan et al., 2025; Ma et al., 2024) (and three-agent (Motwani et al., 2025)) training paradigms. Extending this further, other works such as Qiu et al. (2024); Park et al. (2025); Liao et al. (2025) train multiple models to collaborate to directly collaborate. However, these training schemes can be exceedingly expensive as they require training multiple models. Our framework aims to make multi-agent training more efficient by training only a *single leader agent* without the requirement of training the other models in the team.

**Self-Improvement** Our work is also related to self-improvement. Techniques in this area improve LLM efficacy by iteratively refining given answers. This refinement can come in many forms, such as inference-time mechanisms (Madaan et al., 2023; Wei et al., 2022; Yao et al., 2023; Besta et al., 2024), or trained multi-step reasoning (Zhu et al., 2024; Shao et al., 2024). Of particular relevance to our work is that of Kumar et al. (2024), which trains a model to directly update and refine its own responses. In our method, the leader agent learns to perform a type of macro-update to the agent team responses, while using those responses to synthesize its own refined answer and feedback.

# 3 METHODOLOGY

## 3.1 HIERARCHICAL MULTI-AGENT INFERENCE

**Multi-Agent Team Setup** Our methodology utilizes a hierarchical multi-agent architecture comprising a single leader model $L$ and a team of $K$ off-the-shelf agent models, denoted as $\{a_1, a_2, \ldots, a_K\}$. Although we set $K = 3$ in our experiments, this framework generalizes to any number of agents. Notably, we show that training only the leader model, without requiring that the agent team also be trained, is sufficient to achieve strong collaborative effectiveness and also makes our method considerably more scalable and practical than approaches requiring all agents to be trained.

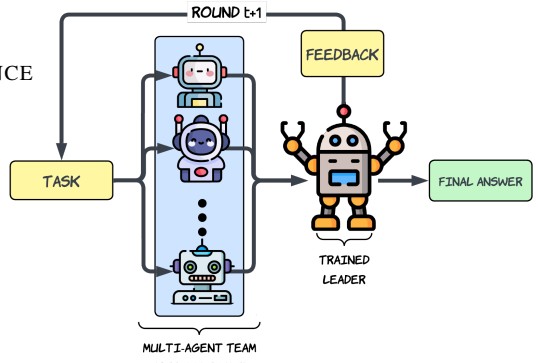

Figure 1: Overview of hierarchical multi-agent inference. A prompt is first processed by a team of $K$ off-the-shelf agents whose intermediate generations are forwarded to the leader (trained using our MLPO). The leader's output is then returned to the agents together with the initial prompt; this repeats for $T$ iterations before producing final answer.

**Multi-Agent Team Inference** The inference process unfolds iteratively over $T$ rounds, as summarised in Figure 1. In the initial round (round 0), each agent independently generates a preliminary solution to a given task input $x$. We denote Agent $i$'s policy by $\pi^{a_i}$ and its 0'th round preliminary solution by $s_i^{(0)} \sim \pi^{a_i}(x)$. The leader model, denoted by $\pi_\theta^L$, then synthesizes these

initial agent responses into its structured output, denoted by $z_L^{(0)}$, i.e.,

$$z_L^{(0)} \sim \pi_\theta^{\mathrm{L}} \left( x, s_1^{(0)}, s_2^{(0)}, \ldots, s_K^{(0)} \right).$$

This output encapsulates detailed reasoning within specific `<think>` tags and presents a consolidated solution within `<answer>` tags. In subsequent rounds ($t = 1, 2, \ldots, T-1$), each agent revises its solution based on the leader's previous output and its own earlier response:

$$s_i^{(t)} \sim \pi^{\mathrm{a}_i} \left( x, s_i^{(t-1)}, z_L^{(t-1)} \right).$$

The leader then integrates these updated agent solutions to refine its reasoning and produce an updated consolidated output:

$$z_L^{(t)} \sim \pi_\theta^{\mathrm{L}} \left( x, s_1^{(t)}, s_2^{(t)}, \ldots, s_K^{(t)} \right).$$

This iterative cycle of feedback and refinement continues until the final round, after which the final solution is extracted from the leader's output.

## 3.2 Multi-Agent Guided Leader Training

Our training approach comprises two phases similar to Guo et al. (2025); Team et al. (2025): Supervised Fine-Tuning (SFT), designed to develop the leader's natural backtracking and self-correction abilities, followed by Group Relative Policy Optimization (GRPO) (Zhu et al., 2024), aimed at enhancing multi-agent collaboration skills.

**Supervised Fine-Tuning** The SFT phase aims to train the leader model to reinforce its aggregation, self-correction and backtracking behaviors, which are crucial skills for effective collaboration and error recovery. To this end, we construct an SFT dataset comprising generations with self-correction and backtracking, inspired by previous approaches explored in recent literature (Qin et al., 2024; Guo et al., 2025; Team et al., 2025); further details can be found in Appendix Section A.

**Multi-Agent Guided Leader Policy Optimization** Following SFT, we employ our variant of Group Relative Policy Optimization (GRPO) (Zhu et al., 2024), which we refer to as Multi-agent guided Leader Policy Optimization (MLPO), to train the leader $\pi_\theta^{\mathrm{L}}$ specifically for multi-agent collaboration. In this phase, the goal is to learn effective aggregation and synthesis strategies. In classical GRPO, each prompt simply comprises the task to be solved (along with any accompanying instructions, such as "let's think step-by-step", etc.). In contrast, for MLPO, each leader's prompt also includes the agents' solutions to the given task. The MLPO loss can therefore be expressed as:

$$\mathcal{J}_{\mathrm{MLPO}}(\theta) = \mathbb{E}_{x \sim P_X, \ \underbrace{\mathbf{s} \sim \boldsymbol{\pi}^{\mathbf{a}}(x)}_{\text{agent responses}}, \ \underbrace{\{o_i\}_{i=1}^G \sim \pi_{\theta_{\mathrm{old}}}^{\mathrm{L}}(x,\mathbf{s})}_{\text{leader response}}} \left[ \frac{1}{\sum_{i=1}^G |o_i|} \sum_{i=1}^G \sum_{t=1}^{|o_i|} \left\{ \min\left( r_{i,t}\hat{A}_{i,t}, \ \mathrm{clip}(r_{i,t}, 1-\varepsilon, 1+\varepsilon)\hat{A}_{i,t} \right) \right\} \right]$$

Here, $r_{i,t} = \frac{\pi_\theta^{\mathrm{L}}(o_{i,t}|x,\mathbf{s},o_{i,<t})}{\pi_{\theta_{\mathrm{old}}}^{\mathrm{L}}(o_{i,t}|x,\mathbf{s},o_{i,<t})}$ is the importance ratio and $\hat{A}_{i,t} = R_i - \mathrm{mean}(\mathbf{R})$ denotes the advantage of the $i$'th response where $R_i$ is the corresponding reward. Moreover, $x$ represents the

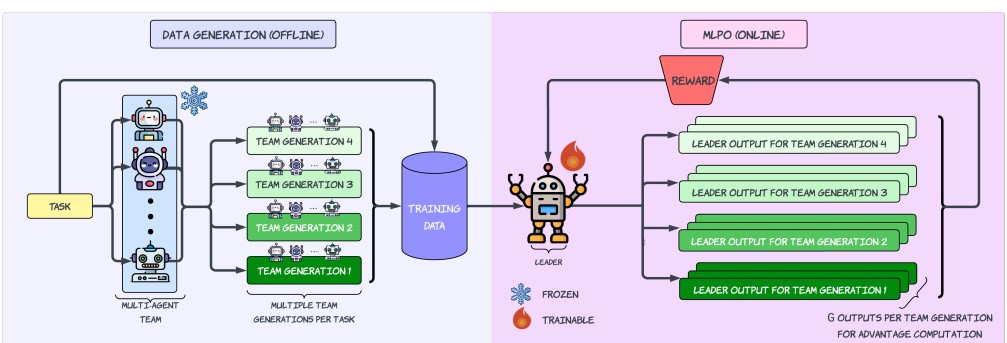

Figure 2: Outline of our Multi-agent guided Leader Policy Optimization (MLPO) pipeline.

task input (e.g. a mathematics problem), $\mathbf{s} = \{s_i^0\}_{i=1}^K$ denotes the set of agent solutions and $o_i$ is a sample from the current leader $\pi_{\theta_{old}}^L$. The hyperparameter $\varepsilon$ denotes the PPO-style clipping threshold (Schulman et al., 2017; Zhu et al., 2024). Note that we have used the modifications to the GRPO loss proposed in Dr. GRPO (Liu et al., 2025) due to its improved training efficiency and stability.

**Training Data Generation**   To generate our MLPO training data, we query each of the $K$ agents to independently generate 4 solutions to each task in the training split, resulting in $4K$ responses per task (as shown in Figure 2). From these responses, we create 4 distinct training prompts per task, each consisting of the original task paired with exactly one response from each agent. In Section 4.4, we demonstrate training with multiple prompts per task significantly improves the leader's accuracy.

Next, we also filter out "easy" tasks, specifically those tasks where at least 75% of the $4K$ agent responses are correct. This filtering directs training attention toward scenarios requiring strong leadership and aggregation skills, which further helps boost the leader's performance, as we demonstrate empirically in Section 4.4. Additionally, while the tasks are randomly shuffled during training, the four prompts corresponding to each task remain grouped. This strategic ordering allows the leader model to observe different agent team responses (and reasoning) to the same task consecutively, and enhances learning efficiency. Figure 6 show that this grouping of prompts by tasks makes training much more stable compared to random ordering of data.

**Zero-Shot and Multi-Agent Inference**   Interestingly, the benefits of our multi-agent guided training extend beyond collaborative settings at inference time. We discovered through our empirical analysis (detailed in Section 4.2.3) that despite the fact that the leader sees *no zero-shot prompts* during training, our MLPO framework significantly enhances the leader's zero-shot inference capabilities as well, even without employing the agent team during inference. This enables us to achieve higher accuracy than models trained with standard single-agent GRPO approaches, at no additional cost during inference. Moreover, when inference-time compute permits, deploying the leader with the agent team further boosts accuracy, outperforming state-of-the-art multi-agent methods. Thus, our framework flexibly provides improved accuracy in both single- and multi-agent inference settings.

## 4 EXPERIMENTAL RESULTS

### 4.1 EXPERIMENTAL SETUP

**Datasets:** We evaluate our hierarchical multi-agent collaboration framework on three benchmarks covering a variety of factual- and reasoning-based tasks: Big-Bench Hard (**BBH**) (Suzgun et al., 2022), **MATH** (Hendrycks et al., 2021), and Massive Multitask Language Understanding (**MMLU**) (Hendrycks et al., 2020).

**Model Configuration:** In our main experiments, we use Qwen2.5 7B Instruct (**Qwen-2.5** (Qwen et al., 2025)) as our leader model, with a heterogeneous agent team comprising Llama 3.1 8B Instruct (**Llama-3.1** (Grattafiori et al., 2024)), Gemma2 9B Instruct (**Gemma-2** (Team et al., 2024)), and Qwen2.5 7B Instruct. In Appendix B.2 we provide results for different choices of agent teams, and in Appendix C we provide results for different leaders (Gemma-2 and Llama-3.1).

**Baselines:** We compare our method MLPO with several baselines. We delineate these baselines into two main categories: training-based and training-free. For training-free baselines, we compare to **Self-Reflection** (Madaan et al., 2023), Multi-Agent Debate (**MAD**) (Du et al., 2023), our pipeline with **untrained leader**, and **zero-shot** inference, which invokes an untrained single LLM once with no multi-agent team support. For training-based baselines, we compare to **ACC-Collab** (Estornell et al., 2024a), **SelectLLM** (Maurya et al., 2024), **SCoRe** (Kumar et al., 2024), a **Deferral leader** trained to always defer to one agent in the team, and **zero-shot GRPO** (Zhu et al., 2024) which invokes a single LLM (trained using standard zero-shot GRPO) once with no multi-agent team support. For all iterative methods, we fix the number of inference rounds as $T = 5$.

### 4.2 MAIN RESULTS

In this section, we present an extensive empirical analysis designed to validate and display the capabilities of our proposed MLPO framework. Specifically, our experiments address the following:

- **Our MLPO vs. Existing Baselines:** We evaluate MLPO against existing approaches, demonstrating how our collaborative reasoning significantly enhances performance.

- **Test-Time Scaling for Improved Performance:** We show how additional computational resources at inference can further amplify the performance gains of our multi-agent team.

- **Multi-agent Guided Training with Zero-Shot Inference**: We demonstrate that our multi-agent guided training pipeline (MLPO) also substantially enhances the leader's zero-shot performance (i.e., without the agent team), surpassing models trained with conventional "single-agent" GRPO methods.

- **Leader-Agent Interaction Dynamics:** We analyze the interaction dynamics between the leader model and its agent team, showing how our MLPO effectively utilizes agent diversity to achieve robust, accurate, and superior outcomes across various task categories.

### 4.2.1 OUR PROPOSED MLPO VS. EXISTING BASELINES

Having outlined our model configuration and selected baselines, we now present the empirical evaluation of our proposed method. Specifically, we systematically investigate our MLPO approach relative to standard single-agent and multi-agent baselines, highlighting the advantages of incorporating collaborative reasoning during both training and inference. Table 1 summarizes our experimental results comparing MLPO with existing baselines.

| TYPE | METHOD | MMLU | BBH | MATH |
|---|---|---|---|---|
| TRAINING-FREE | ZERO-SHOT* | $0.734_{\pm0.006}$ | $0.733_{\pm0.010}$ | $0.666_{\pm0.002}$ |
| | SELF-REFLECT (MADAAN ET AL., 2023) | $0.746_{\pm0.003}$ | $0.766_{\pm0.011}$ | $0.681_{\pm0.003}$ |
| | MAD (DIV) (DU ET AL., 2023) | $0.771_{\pm0.002}$ | $0.785_{\pm0.002}$ | $0.653_{\pm0.009}$ |
| | MAD (HOM) (DU ET AL., 2023) | $0.759_{\pm0.003}$ | $0.799_{\pm0.016}$ | $0.720_{\pm0.009}$ |
| | UNTRAINED LEADER | $0.731_{\pm0.008}$ | $0.764_{\pm0.007}$ | $0.697_{\pm0.007}$ |
| TRAINING-BASED | ACC-COLLAB (ESTORNELL ET AL., 2024A) | $0.761_{\pm0.004}$ | $0.802_{\pm0.003}$ | $0.698_{\pm0.008}$ |
| | SELECTLLM* (MAURYA ET AL., 2024) | $0.770_{\pm0.001}$ | $0.768_{\pm0.001}$ | $0.651_{\pm0.001}$ |
| | SCORE (KUMAR ET AL., 2024) | $0.752_{\pm0.005}$ | $0.828_{\pm0.004}$ | $0.721_{\pm0.004}$ |
| | DEFERRAL LEADER | $0.750_{\pm0.004}$ | $0.775_{\pm0.007}$ | $0.720_{\pm0.004}$ |
| | ZERO-SHOT GRPO* (ZHU ET AL., 2024) | $0.742_{\pm0.004}$ | $0.791_{\pm0.008}$ | $0.712_{\pm0.005}$ |
| | SELF-REFLECT GRPO ((ZHU ET AL., 2024) + (MADAAN ET AL., 2023)) | $0.747_{\pm0.003}$ | $0.814_{\pm0.006}$ | $0.718_{\pm0.006}$ |
| OUR RESULTS | SFT + MLPO LEADER* (ZERO-SHOT ACC W/O TEAM) | $0.757_{\pm0.007}$ | $0.855_{\pm0.008}$ | $0.729_{\pm0.008}$ |
| | SFT LEADER | $0.741_{\pm0.008}$ | $0.762_{\pm0.008}$ | $0.713_{\pm0.007}$ |
| | MLPO LEADER | $0.759_{\pm0.006}$ | $0.865_{\pm0.005}$ | $0.750_{\pm0.009}$ |
| | SFT + MLPO LEADER | $\mathbf{0.782}_{\pm0.006}$ | $\mathbf{0.882}_{\pm0.005}$ | $\mathbf{0.762}_{\pm0.005}$ |

Table 1: Accuracy of each method on MMLU, BBH, and MATH benchmarks, with $\pm 2$ standard errors. Here, * denotes methods not using iterative inference.

Our method substantially outperforms both single-agent training baselines (e.g., SCoRe (Kumar et al., 2024), GRPO (Zhu et al., 2024)) and multi-agent training baselines (e.g., ACC-Collab (Estornell et al., 2024a), SelectLLM (Maurya et al., 2024), Deferral Leader). This performance gain stems from the fact that, during both training and inference, the leader is guided by a diverse set of strategies and solutions proposed by the agent team. In contrast, single-agent approaches operate in isolation, lacking access to this breadth of insight, as they must rely on their own solutions.

Other than demonstrating superior performance, these results also help ensure that our leader develops decision-making capabilities beyond replicating or selecting agent responses. For this, we compare to SelectLLM (Kumar et al., 2024), which identifies the optimal agent subset based solely on the input question, and Deferral Leader, which selects the best response given both the question and agent outputs. Our trained leader consistently outperforms these baselines across all evaluated domains, providing evidence of its capability to formulate independent strategies beyond mere imitation. This conclusion is further corroborated by Figure 5, where our leader maintains significantly higher accuracy even in scenarios where all individual agents fail.

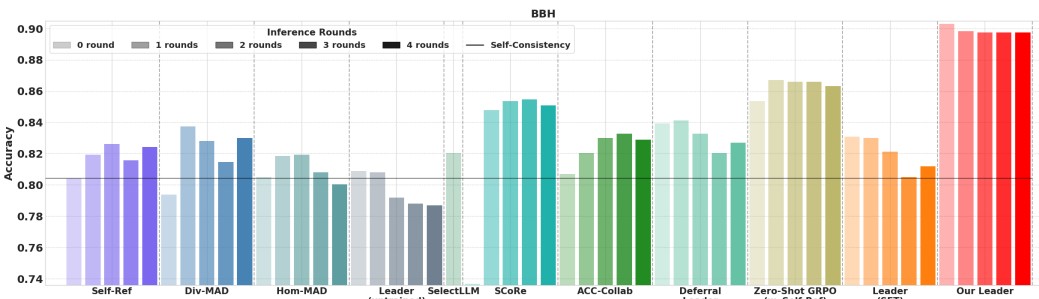

Figure 3: Majority vote performance when each method can use at most 40 total LLM generation samples. For iterative methods, we only consume the final round answers for majority vote. Traditional self-consistency (Wang et al., 2022) is shown as horizontal line. The leader model trained with SFT + MLPO is denoted as Our Leader.

### 4.2.2 TEST-TIME SCALING FOR IMPROVED PERFORMANCE

A natural question to ask is whether our improvements observed in the previous Section are merely a consequence of our method being allowed to use more generations during inference. To address this concern, we investigate performance when all methods are allotted the same amount of inference time compute, and explore how additional inference-time compute can be leveraged to further improve the performance of models trained with MLPO under a fixed inference budget.

In particular, prior work has shown that majority voting over multiple generations, can enhance model accuracy (Wang et al., 2022). For fair comparison, we constrain all methods (including iterative and multi-agent baselines) to a maximum of 40 total LLM generations at inference time. For iterative methods, we apply majority voting only over responses from the final inference round.

Figures 3 reports accuracy under this consistent sample budget with majority voting for BBH, results for MMLU and MATH are similar and shown in Figure 7. These results show that our method achieves superior performance across all three datasets even when inference-time compute is scaled.

Our method may benefit particularly from parallel scaling due to having two distinct sources of diversity: variability in responses from the agent team and stochasticity in the leader's own generations. This dual source of response diversity could help explain why our pipeline yields higher majority-vote accuracy under fixed sample budgets, compared to baselines that lack these mechanisms.

### 4.2.3 LEADER'S ENHANCED ZERO-SHOT CAPABILITIES (NO AGENT TEAM)

Remarkably, our multi-agent trained leader also exhibits superior performance even during zero-shot inference (without multi-agent assistance), outperforming models trained with standard SFT + GRPO pipelines, hence incurring **no additional inference cost**; see Figure 4.

This finding suggests that MLPO enhances "*knowledge acquisition*" during training, potentially through improved exploration of alternative solutions proposed by the agent team. The key difference between our multi-agent guided training framework MLPO, and standard single-agent training

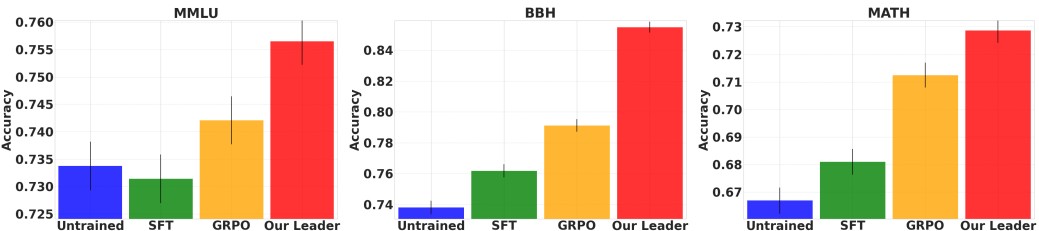

Figure 4: Zeroshot performance of Qwen when untrained, SFTed or GRPOed on zero-shot data, and trained via our MLPO pipeline (Our Leader). The error-bars show $\pm 2$ standard errors.

frameworks such as GRPO, lies in the training prompts: the former includes agent solutions alongside the question, while the latter contains only the question and instructions. The inclusion of diverse agent responses appears to create a richer training signal that improves the leader's reasoning capabilities even when later deployed in isolation.

An astute reader may ask if this enhanced "knowledge acquisition" is primarily responsible for the improved performance of our multi-agent system. In other words, *could it be that even when deployed with the multi-agent team, the trained leader is simply ignoring the agent responses at inference time and using its acquired knowledge to drive the accuracy gains*? Our results in Table 1 show that this is not the case, since the leader's accuracy improves even further when it is deployed with the multi-agent team at inference time (as compared to its zero-shot inference performance). Not only has the leader acquired knowledge, but also utilizes its implicit evaluation and aggregation skills to synthesize a higher-quality final solution when given access to the agent team's responses.

### 4.3 AGENT TEAM AND LEADER DYNAMICS

Next, dive deeper into the dynamics between the leader and the agent team. The bulk of this investigation and discussion is provided in Appendix B.2 and we outline our main findings here.

**Leader Robustness to Incorrect Agent Solution** An ideal leader is one that can transcend the abilities of any one model, or even the team as a whole. Such behavior necessitates that the leader can reliably override incorrect answers provided by the agent team. Figure 5 shows that the leader trained with MLPO is far more able to override incorrect solutions proposed by the team compared with other types of leaders. Additionally, Appendix Figure 8 shows that leader performance dominates the performance of even the best model in the team.

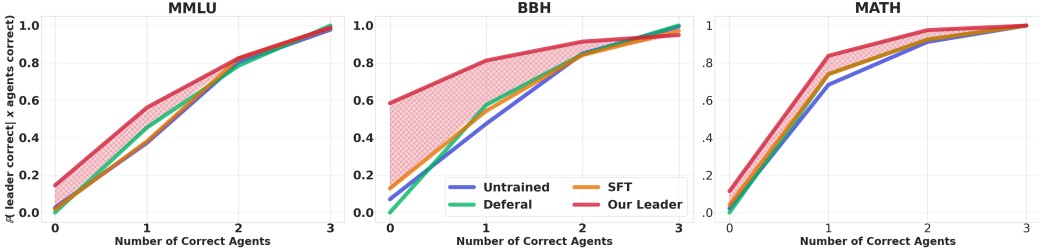

Figure 5: Leader performance conditioned on the number of correct agents in the team. Shaded regions represent the degree to which our leader outperforms the next best method.

**Multi-Round Training (MLPO+)** Recall that for efficiency, MLPO leverages only a single round of agent-team generations during training. While we find that this is sufficient out perform existing methods, we also investigate a version of MLPO which makes use of an additional training phase using multi-round generations (MLPO+) which further enhances performance (Table 2. See Appendix Section B.2 and Figure 9 for more details.

|  | MMLU | BBH | MATH |
|---|---|---|---|
| MLPO | $0.782_{\pm 0.006}$ | $0.882_{\pm 0.005}$ | $0.762_{\pm 0.005}$ |
| MLPO+ | $0.792_{\pm 0.005}$ | $0.920_{\pm 0.004}$ | $0.771_{\pm 0.005}$ |

Table 2: Results of a leader trained with MLPO and MLPO+ after 5 rounds of inference

We also investigate alternative model choices for the agent team and find that a diverse team (e.g., Qwen, Llama, Gemma) results in the best overall performance (Appendix Figure 12).

### 4.4 BEST PRACTICES FOR MULTI-AGENT GUIDED TRAINING

We conclude by highlighting key considerations in our multi-agent guided training paradigm that helped guide the design of our method. For the full set of findings see Appendix B.3, here we provide an brief overview of some of the key points.

**Solution Set Ordering**   First, we discuss the ordering in which the leader is exposed to the different sets of solutions. For our main experiments, we sample four sets of agent solutions for each task $x$. In Figure 6 we see difference in reward curves for training with shuffled solution sets, and grouped solution sets (i.e., the leader sees all four solution sets in a single batch). Grouping solutions by task makes training more stable and results in higher reward for the trained leader; indicating that there is a benefit during training of seeing multiple similar (but different) examples in a single batch.

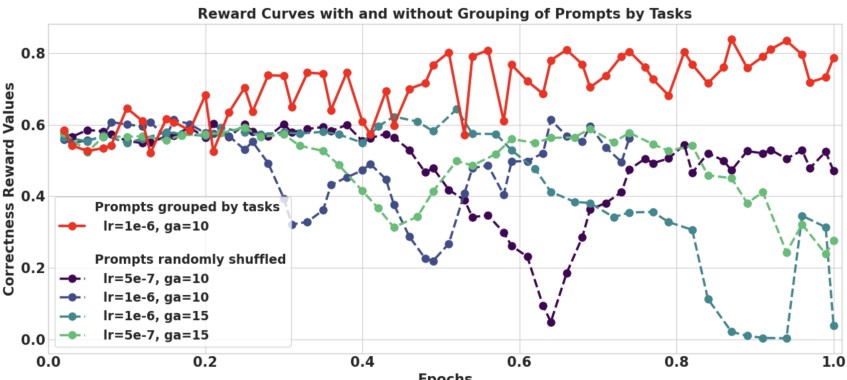

Figure 6: Reward curves for training with grouped solution sets vs randomly shuffled solution sets. Learning rate and gradient accumulation steps are denoted respectively as *lr* and *ga*.

In Appendix B.3 we also conduct an ablation on the number of samples per task to include in the training data and find that 4 samples per task is ideal (Appendix Table 9. Further, we also provide an ablation on our choice to filter "easy" questions out of the training data and find that it does indeed improve performance (Appendix Table 10). We also show that RL-based methods perform better when trained with more diverse alternative solutions (Appendix Table 8).

**Alterative Choice of Leader**   Lastly we remark on our choice of model for the trained leader. Throughout the main body, we present results when using Qwen-2.5 as the leader. In Appendix C we present our main set of results for alterative leaders. We observe similar results, utilizing our multi-agent guided training results in the highest efficacy, although we observe less performance increase using Gemma-2 and Llama-3.1 as the leader. This suggested that the most effective leaders may be models capable of more advanced reasoning, such as Qwen-2.5. This matches with intuition as an effective leader must be skilled at reasoning through the agent responses.

## 5   CONCLUSION

We introduce a hierarchical multi-agent framework leveraging our proposed Multi-Agent Guided Leader Optimization (MLPO) method, specifically designed to enhance the collaborative reasoning capabilities of large language models (LLMs). By explicitly training only a single leader model, our approach introduces novel multi-agent guided training and inference techniques, effectively integrating diverse insights from a team of untrained peer agents. Extensive empirical results on challenging benchmarks demonstrate significant performance improvements over single-agent and other multi-agent baselines.

Nevertheless, our method also has certain limitations. These include increased context lengths required both during training and inference, higher computational demands at inference time for optimal performance, and reduced parallelizability due to sequential leader-agent interactions. Further, MLPO relies on a central leader model, and is more sensitive to the leader's efficacy (compared with other multi-agent approaches that do not utilize a central agent); as such, we recommend using a model with comparable strength to the agent team. Addressing these limitations could involve exploring strategies like selective agent querying and caching to reduce training/inference load as well as different agent team combinations to reduce the number of rounds needed. Despite these constraints, our framework remains highly adaptable. It supports both parallel and sequential scaling strategies as needed, and notably, achieves strong performance even when deployed without the agent team at inference time.

**Reproducibility Statement:** We outline in detail our methodology in Section 3 of the main body and provide more complete details on implementation and experimental design in Section B of the Appendix. We provide a full description of the configurations used for each baseline method in D. We provide example prompts and templates used in Section E. All code will be made public upon publication of our work.

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

APPENDIX

# A    ADDITIONAL EXPERIMENTAL DETAILS

## A.1    SUPERVISED FINE-TUNING

As mentioned in the main body, we deploy an initial SFT step prior to RL-training. The SFT phase aims to train the leader model to reinforce its aggregation, self-correction and backtracking behaviors, which are crucial skills for effective collaboration and error recovery. To this end, we construct a new SFT dataset comprising generations with self-correction and backtracking, inspired by previous approaches explored in recent literature (Qin et al., 2024; Guo et al., 2025; Team et al., 2025). To achieve this, we first randomly select one correct and one incorrect leader response from the 16 leader generations. Next, an untrained leader model is prompted with these selected responses to generate a "*backtracked solution*" that mimics natural reasoning patterns, i.e. beginning with plausible but incorrect reasoning, then naturally self-correcting to arrive at the correct answer. The prompts explicitly encourage natural self-correction phrases such as "Wait, that doesn't seem right" or "Let me reconsider this approach". For tasks where all 16 responses are correct, we directly include the correct solutions without artificial backtracking. The leader model is then fine-tuned on this dataset of backtracked and correct solutions, aiming to enhance its ability to naturally self-correct and recover from reasoning errors.

## A.2    MULTI-ROUND TRAINING WITH MLPO+

Initially, our pipeline trains the leader solely on agent-team responses from round 0, i.e., responses generated before any interaction between the leader and the agent team has occurred. This setup enables efficient offline training, as round-0 responses can be precomputed prior to leader optimization. While this training scheme is already effective, it does not leverage the full distribution of team responses that arise at inference.

To address this, we also consider a multi-round extension of MLPO, which we call MLPO+, that incorporates an additional training phase using team responses from later rounds ($t > 0$). These responses resemble more closely those seen during inference, as they are shaped by interaction with the leader. To operationalise MLPO+ efficiently, we treat the multi-round training as a continuation phase that follows the initial MLPO training. After the leader is trained using round-0 responses, it is deployed to interact with the agent team over multiple rounds. These interactions are then recorded and used to construct a new training set. Importantly, this additional data can be generated offline, just like the round-0 responses, introducing little extra overhead.

Similar to the first round of training, we first filter out "easy" tasks, however we now set the accuracy threshold much higher (those where the leader and team achieve over 50% accuracy after two rounds). For the remaining tasks, we construct a new dataset consisting of two sets of agent responses from rounds 0, 1, and 2. The leader is then further optimized on this dataset using the same MLPO loss. This setup allows the leader to refine its strategy based on more realistic team behaviors that emerge during iterative interaction.

### A.2.1    MULTI-AGENT TEAM INFERENCE

Recall that at inference time the leader observes the task $x$ and the agent team's responses $s_1^{(t)}, s_2^{(t)}, \ldots, s_K^{(t)}$ and provides its response

$$z_L^{(t)} \sim \pi_\theta^{\mathrm{L}} \left( x, s_1^{(t)}, s_2^{(t)}, \ldots, s_K^{(t)} \right).$$

Importantly, the leader does not see its own response from

# B    ADDITIONAL EXPERIMENTS

## B.1    TEST-TIME SCALING FOR IMPROVED PERFORMANCE

**Sample Normalized Majority Vote**: Figure 7 shows majority vote results for all methods on each of the three baselines when given a budget of 40 total samples. We observe that that similar to BBH

(presented in the main body) that our method results in a leader that has superior performance to all baseline methods.

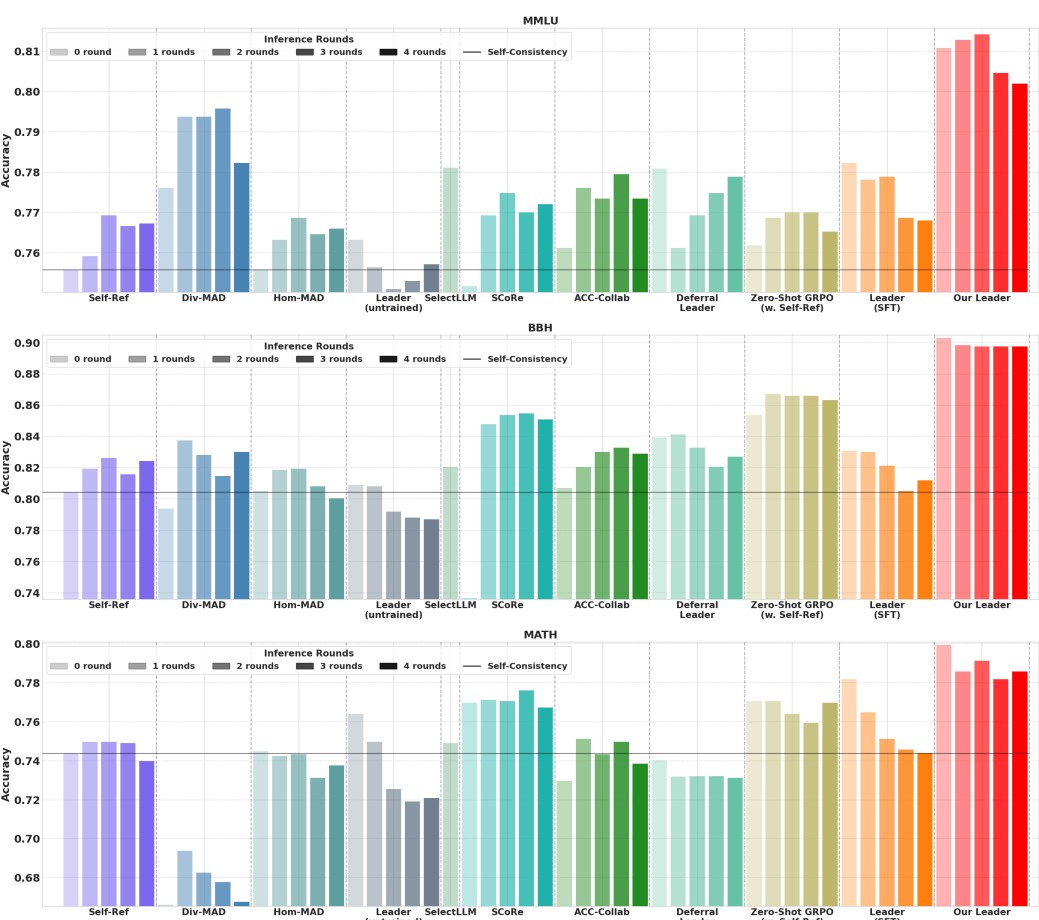

Figure 7: Majority vote performance when each method can use at most 40 total LLM generation samples. For iterative methods, we only consume the final round answers for majority vote. Traditional self-consistency (Wang et al., 2022) is shown as horizontal line. The leader model trained with SFT + MLPO is denoted as Our Leader.

**Token Normalized Majority Vote:** In addition to the sample normalized majority vote results presented in the main body, we also examine the efficacy of each method when normalizing by inference-time token usage. For each method we use two rounds of inference (average accuracy of each method tends to plateau after two rounds) and generate $k$ answers until the token budget is hit. If generating one answer exceeds the token budget (i.e., $k = 0$) the answer is automatically counted as wrong.

In Table 3 we see that for or all larger budgets (10k and above), the MLPO Leader consistently beats all baselines. Additionally, for these larger token budget ,the MLPO leader without the team consistently beats a model trained only with GRPO. In cases where token generations are limited, approaches such as GRPO or SCoRe may be preferable, but in cases of larger token budgets MLPO is the superior method.

## B.2   LEADER AND AGENT TEAM INTERACTION DYNAMIC

**Leader vs Agent Team across Category and Difficulty**    An ideal leader can leverage the unique strengths of the multi-agent team to produce solutions which are better than any one model could alone. Figure 8 shows the performance of our trained leader and each individual agent in the leader's team. Within these plots, we see two interesting observations.

| | MMLU | | | | | | | |
|---|---|---|---|---|---|---|---|---|
| Token Budget | Self-Ref | MAD (hom) | MAD (div) | Untrained Leader | GRPO | SCoRe | MLPO Leader (Ours) | MLPO No Team (Ours) |
| 2,500 | 0.750 | 0.414 | 0.738 | 0.364 | 0.749 | **0.757** | 0.074 | 0.704 |
| 5,000 | 0.762 | 0.745 | **0.786** | 0.732 | 0.766 | 0.767 | 0.761 | 0.763 |
| 7,500 | 0.759 | 0.752 | **0.790** | 0.748 | 0.765 | 0.767 | 0.786 | 0.769 |
| 10,000 | 0.763 | 0.757 | 0.788 | 0.755 | 0.767 | 0.767 | **0.793** | 0.771 |
| 15,000 | 0.763 | 0.760 | 0.797 | 0.758 | 0.769 | 0.767 | **0.805** | 0.773 |
| 20,000 | 0.764 | 0.759 | 0.797 | 0.765 | 0.769 | 0.767 | **0.818** | 0.773 |
| | BBH | | | | | | | |
| 2,500 | 0.773 | 0.554 | 0.757 | 0.280 | 0.795 | **0.828** | 0.026 | 0.720 |
| 5,000 | 0.803 | 0.797 | 0.815 | 0.734 | **0.842** | 0.841 | 0.749 | 0.832 |
| 7,500 | 0.808 | 0.811 | 0.823 | 0.754 | 0.852 | 0.840 | **0.864** | 0.863 |
| 10,000 | 0.817 | 0.819 | 0.827 | 0.777 | 0.858 | 0.844 | **0.890** | 0.870 |
| 15,000 | 0.823 | 0.826 | 0.831 | 0.795 | 0.858 | 0.848 | **0.896** | 0.872 |
| 20,000 | 0.824 | 0.827 | 0.834 | 0.811 | 0.860 | 0.848 | **0.900** | 0.874 |
| | MATH | | | | | | | |
| 2,500 | 0.705 | 0.461 | 0.534 | 0.008 | 0.656 | **0.730** | 0.000 | 0.601 |
| 5,000 | 0.726 | 0.691 | 0.685 | 0.494 | 0.735 | **0.745** | 0.394 | 0.731 |
| 7,500 | 0.737 | 0.724 | 0.696 | 0.616 | **0.761** | 0.759 | 0.648 | 0.756 |
| 10,000 | 0.740 | 0.731 | 0.698 | 0.664 | 0.767 | 0.765 | **0.778** | 0.771 |
| 15,000 | 0.746 | 0.742 | 0.698 | 0.698 | 0.770 | 0.768 | **0.783** | 0.778 |
| 20,000 | 0.749 | 0.746 | 0.698 | 0.714 | 0.771 | 0.769 | **0.785** | 0.779 |

Table 3: Majority vote accuracy across token budgets for each dataset.

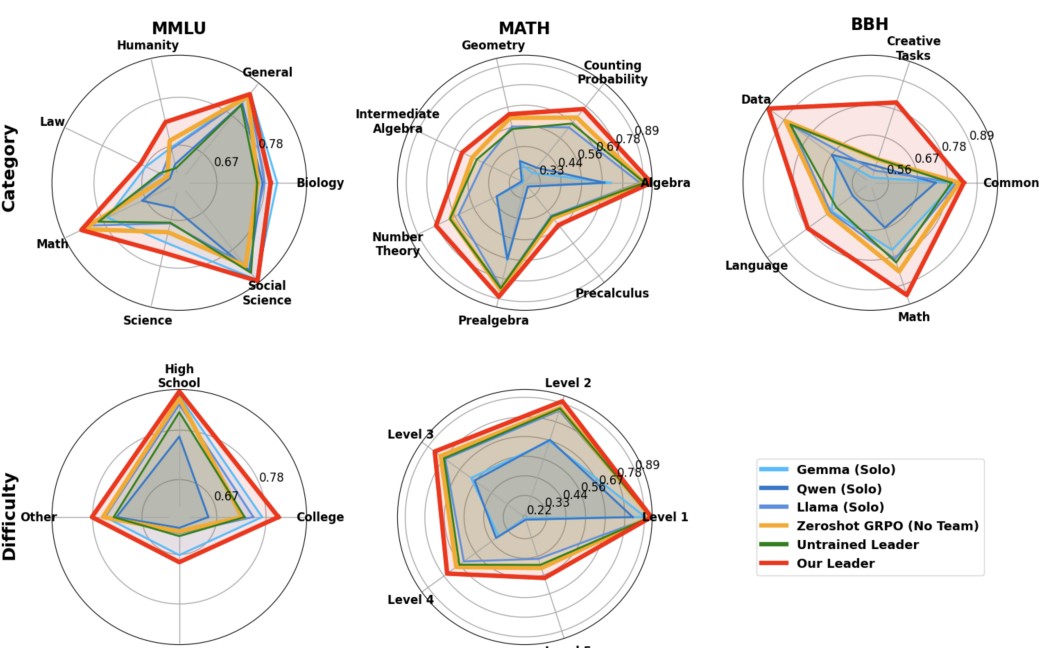

Figure 8: Our leader trained with MLPO, compared with an untrained leader, to zero-shot GRPO, and individual team performance, per category (top) and difficulty level (bottom) on MMLU (left) MATH (center) and BBH (right). Accuracy is reported after 5 rounds of inference. Note that BBH does not have difficulty gradations.

First, the multi-agent pipeline with an untrained leader (green) struggles to surpass each of the individual agents (blue), both per-category and per-difficulty across each of the benchmarks. In contrast, the multi-agent team with an MLPO trained leader can surpass the performance of each agent as well as the untrained leader in nearly all categories and difficulty (occasional ties with the best agent). Second, the model trained with zero-shot GRPO (orange) also struggles to dominate the untrained leader, as well as each agent, over categories and difficulties. The per-category and

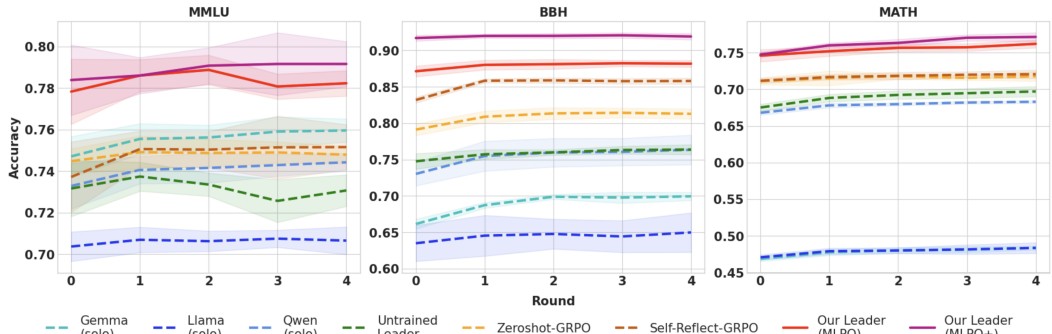

Figure 9: Performance over multiple rounds of interaction. Shaded region represents $\pm 2$ standard errors.

per-difficulty success of the zero-shot model is far more task-dependent than that of our trained leader. In particular, when examining MMLU, we see that the untrained agents or the untrained leader beats the zero-shot GRPO model on 4 our of 7 categories (Science, Social Science, Biology, and Law) and 2 out of 4 difficulties (professional and college). However, our trained leader dominates in all difficulties and 6 out of 7 categories (losing only on Biology), highlighting that the leader trained via our MLPO framework successfully acquires a balanced and robust skill set. MLPO consistently outperforms both the (trained) individual agents as well as the team with an untrained leader across almost all categories and difficulty levels.

**Per-Round Improvement** Next, we examine the accuracy improvements achieved through iterative rounds of discussion between the leader and the agent team, as well as the self-reflection performance of individual agents (solo). Figure 9 (and previously shown Figure 3) presents the accuracy progression for each method across these multiple rounds. Notably, even though our leader model is only trained on initial round data (round 0), we consistently observe slight accuracy improvements in subsequent rounds. One possible explanation for this improvement is that in later rounds, agents benefit from additional information provided by the leader's earlier responses, enabling them to refine their own answers. This, in turn, may further assist the leader model in correcting its previous errors. However, further investigation is required to fully understand this phenomenon.

**Multi-Round Training with MLPO+** We now discuss a complementary phenomenon: the ability to further improve leader performance through additional training. We evaluate the performance of MLPO+, which fine-tunes the leader on agent-team responses from later rounds ($t > 0$), and find that it yields consistent improvements in leader performance.

The performance boost appears to stem from two complementary factors: first, the leader is trained on responses that better reflect those it will encounter during deployment, and second, it learns to better synthesize responses from agents that have already been shaped by its prior guidance. Notably, these gains come with minimal additional computational cost, as the extra training data can still be collected offline. Overall, these results highlight the value of iterative supervision: exposing the leader to more realistic interactions and more divers solutions during training increases the leader's efficacy.

|       | MMLU              | BBH               | MATH              |
|-------|-------------------|-------------------|-------------------|
| MLPO  | $0.782_{\pm 0.006}$ | $0.882_{\pm 0.005}$ | $0.762_{\pm 0.005}$ |
| MLPO+ | $0.792_{\pm 0.005}$ | $0.920_{\pm 0.004}$ | $0.771_{\pm 0.005}$ |

Table 4: Results of a leader trained with MLPO and MLPO+ after 5 rounds of inference

**Leader Robustness to Agent Team's Ability** An effective leader should consistently produce correct solutions, by leveraging accurate responses from agents when available, and overriding incorrect ones when necessary. In Figure 10, we compare the performance of various leaders as a

function of agent correctness. To ensure a fair comparison, we report results only for round 0, where agent responses are not influenced by the leader. While all leaders perform well when all agents are correct, their accuracy drops substantially when agent correctness is low. Notably, our trained leader shows strong performance particularly when only few agents are correct, highlighting our method's robustness to low team performance. This effect is especially pronounced on the BBH dataset.

One possible explanation for this improved robustness could be better knowledge acquisition by the leader model during MLPO, enabling it to independently generate correct answers without relying solely on the agent solutions at inference time. This hypothesis aligns with the observed increase in zero-shot accuracy as previously mentioned in Figure 4. However, knowledge acquisition alone does not fully explain our leader's performance. Specifically, we observe additional improvements when correct agent responses are available at inference time. This indicates that the leader has learned not only to independently solve problems but also to effectively evaluate the quality of agent inputs, discerning when to trust their advice and when to override incorrect team responses.

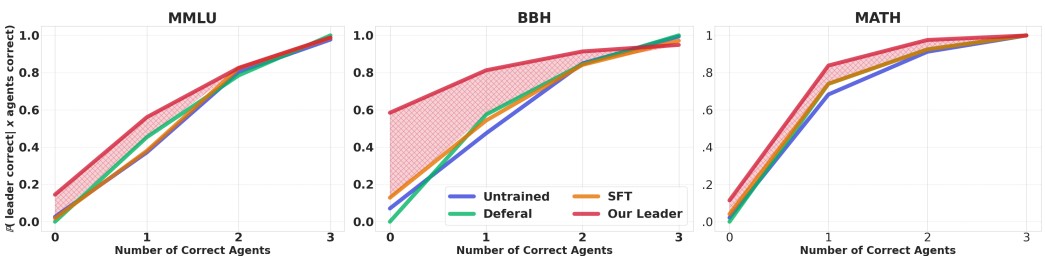

Figure 10: Leader performance conditioned on the number of correct agents in the team. Shaded regions represent the degree to which our leader outperforms the next best method.

**Leader's Utilization of Agent Information**  To better understand the mechanisms behind our trained leader's superior performance, we analyze how it utilizes the information provided by the agent team. We conducted ablation experiments, selectively providing either agent reasoning, final answers, or both, to the leader during inference. The experiments are summarized in Figure 11.

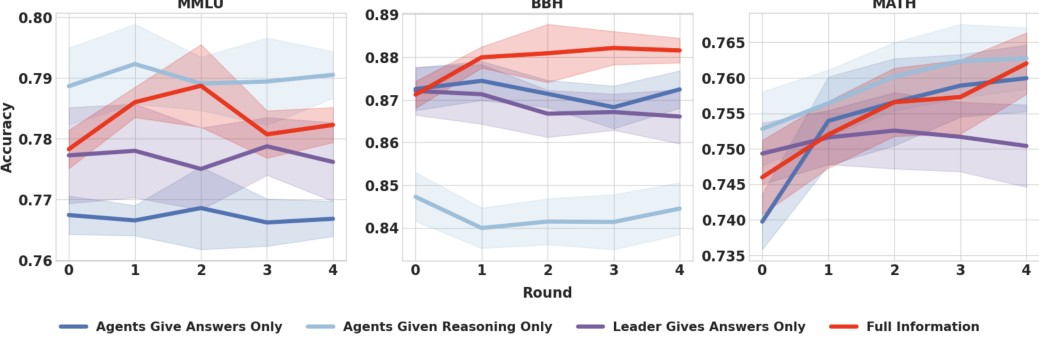

Figure 11: Leader performance over rounds of inference, varying by which information is shared between the agent team and the leader. Shaded region represents $\pm 2$ standard errors.

Our findings consistently indicate that, in most scenarios, providing the leader with both the agents' reasoning processes and their final answers achieves the highest performance. This suggests that the leader effectively leverages the complete context provided by the agents, benefiting from both their detailed thought processes and their concluded solutions.

Interestingly, in the majority of cases, reasoning-only input achieves better performance than final-answer-only input. This implies that the leader primarily derives value from understanding the agents' underlying reasoning rather than merely aggregating their final decisions. However, notable exceptions exist: for the BBH dataset, final-answer-only surpasses reasoning-only performance, although providing both components remains optimal. When deploying leaders trained with MLPO, utilizing agent full responses generally elicits the best leader performance.

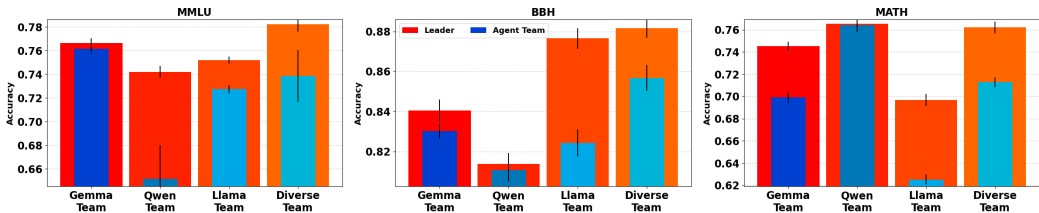

Figure 12: Leader accuracy (warm colors) and average team accuracy (cool colors) after 5 rounds of interaction.

These observations indicate that the relative importance of reasoning versus final answers depends critically on both the dataset characteristics and the specific training regime. Including final answers may sometimes bias the leader towards incorrect conclusions, whereas reasoning-only input often encourages independent verification and robust inference. Nonetheless, overall, providing both reasoning and final answers to the leader consistently yields optimal or near-optimal results across diverse scenarios.

**Team Composition**  To better understand the interaction dynamics between the leader and its team, we evaluate leader performance under different choices of agent teams; namely the diverse team used in our main results (Qwen-2.5, Llama-3.1, and Gemma-2), and homogeneous teams consisting of 3 copies of each model (e.g., 3 copies of Gemma-2). For each team configuration, a leader is trained specifically for that team. Figure 12 reports both the average leader accuracy and the average team accuracy after five rounds of interaction.

Two key observations emerge from these results. First, leaders paired with homogeneous teams exhibit more task-dependent performance. For example, on BBH dataset, the leader paired with Llama team achieves highest accuracy among all homogeneous teams, while on MATH, it achieves the lowest. However, the leader paired with the heterogeneous team consistently matches or exceeds the performance of leaders paired with homogeneous teams. This highlights the benefit of deploying MLPO with a diverse agent team. Second, the gap between leader and team performance can be substantial, even after the team has had extensive interaction with the leader. This underscores the leader's critical role in providing high-quality solutions even when the underlying team struggles.

**Training with Different Team Sizes**  In addition to studying the effects of team composition, we also examine the effect of training with teams of differing sizes. We focus on homogeneous teams consisting of Gemma, Qwen, or Llama, since altering the number of agents with diverse teams introduces the confounder of which model to remove or add. We study teams of size 2, 3, and 4.

In Table 5 we see that Performance is fairly stable across team sizes. Moreover, there is a general trend that using more models (4 vs 3 or 2) is better. However we would not expect this to hold as additional agents continue to be included. Lastly, a diverse team remains the best choice as it either beats, or ties with, each of the homogeneous teams, even when that team has an additional agent.

**Generalization to Different Agent Teams**  To test how well trained leaders can generalize to other team compositions we deploy a Qwen leader, trained on a heterogeneous team, with different homogeneous teams of size 3. In Table 6 we see that the trained leader exhibits a strong ability to generalize from a heterogeneous team to each of the homogeneous teams. Even when paired with teams that the leader was not trained on, the leader consistently outperforms both the untrained leader and a model trained with GRPO. In some cases, the leader trained on a heterogeneous team, paired with a homogeneous team at inference time, can outperform a leader trained on that homogeneous team (Figure 12), e.g., the Qwen team on BBH. We believe this gives more evidence for our hypothesis that exposing the leader to diverse solutions at inference is a key element to the leader's success (solutions will be naturally more diverse with a heterogeneous team).

| MMLU | 2 | 3 | 4 |
|------|-----|-----|-----|
| Gemma Team | 0.759 | 0.765 | 0.776 |
| Qwen Team | 0.744 | 0.741 | 0.745 |
| Llama Team | 0.740 | 0.752 | 0.758 |
| Diverse Team | – | 0.782 | – |

| BBH | 2 | 3 | 4 |
|------|-----|-----|-----|
| Gemma Team | 0.833 | 0.840 | 0.842 |
| Qwen Team | 0.839 | 0.812 | 0.842 |
| Llama Team | 0.861 | 0.878 | 0.869 |
| Diverse Team | – | 0.882 | – |

| MATH | 2 | 3 | 4 |
|------|-----|-----|-----|
| Gemma Team | 0.730 | 0.744 | 0.746 |
| Qwen Team | 0.741 | 0.762 | 0.765 |
| Llama Team | 0.705 | 0.697 | 0.690 |
| Diverse Team | – | 0.762 | – |

Table 5: Performance of a Qwen leader when trained with teams of differing sizes.

| Agent team | MMLU | BBH | MATH |
|------------|------|-----|------|
| Qwen + Gemma + Llama (seen) | **0.782** | **0.882** | **0.762** |
| 3x Qwen (unseen) | 0.760 | 0.858 | 0.761 |
| 3x Gemma (unseen) | 0.762 | 0.868 | 0.701 |
| 3x Llama (unseen) | 0.745 | 0.862 | 0.680 |
| No Team (unseen) | 0.757 | 0.855 | 0.729 |
| Untrained Leader (baseline) | 0.731 | 0.764 | 0.697 |
| GRPO (baseline) | 0.742 | 0.791 | 0.712 |

Table 6: Performance of a Qwen Leader trained on a heterogeneous team (Qwen + Llama + Gemma) and deployed on homogeneous teams after 5 rounds of inference. Highest and second highest performing methods are bolded and underlined respectivly.

**Size of the Agent Team**    To evaluate whether MLPO remains effective as the number of untrained agents increases, deploy our trained leader (trained on a heterogeneous team of 3 games) with teams of size $K$, where $0 \leq K \leq 6$. To isolate the effect of team *size* rather than team *composition*, we focus on homogeneous teams of size $K$. We compare our trained leader against the untrained leader under identical settings.

As shown in Table 7, we observe two consistent trends across MMLU, BBH, and MATH, . First, the generalization trends observed in Table 6 continue to hold, as the trained leader is quite robust to the choice of team size; its performance varies only mildly as $K$ increases, and in almost all cases it outperforms the untrained leader for every $K$. Second, when the agent team is particularly weak (e.g., Llama-3.1-8B teams on MATH), adding more agents can *hurt* performance, and the best choice is sometimes $K = 0$; this reinforces our finding that heterogeneous teams provide the strongest overall gains (outlined in Fig. 12).

## B.3    BEST PRACTICES FOR MULTI-AGENT GUIDED TRAINING

We conclude by highlighting key considerations in our multi-agent guided training paradigm that helped guide the design of our method. Recall that during training, the leader observes the responses generated by the agent team before producing its own answer. Thus, the leader is trained on prompt-response pairs of the form (`task + agent_responses`, `leader_response`), in contrast to standard single-agent training, which relies solely on the task input. This distinction gives rise to two natural questions:

| Agent Team Size ($K$) | 0 (No Team) | 1 | 2 | 3 | 4 | 5 | 6 |
|---|---|---|---|---|---|---|---|
| **MMLU** | | | | | | | |
| Qwen (trained/untrained) | .757/.746 | .757/.698 | .758/.723 | .760/.720 | .761/.722 | .752/.699 | .748/.681 |
| Gemma (trained/untrained) | .757/.746 | .760/.741 | .762/.751 | .762/.757 | .759/.756 | .757/.750 | .750/.743 |
| Llama (trained/untrained) | .757/.746 | .752/.729 | .750/.730 | .745/.738 | .743/.727 | .741/.732 | .737/.719 |
| **BBH** | | | | | | | |
| Qwen (trained/untrained) | .855/.766 | .856/.753 | .856/.750 | .858/.759 | .856/.760 | .853/.763 | .849/.747 |
| Gemma (trained/untrained) | .855/.766 | .858/.751 | .866/.751 | .868/.750 | .865/.750 | .859/.755 | .843/.741 |
| Llama (trained/untrained) | .855/.766 | .860/.755 | .860/.751 | .862/.755 | .857/.756 | .840/.765 | .839/.763 |
| **MATH** | | | | | | | |
| Qwen (trained/untrained) | .729/.681 | .758/.755 | .756/.751 | .761/.759 | .755/.754 | .758/.762 | .756/.754 |
| Gemma (trained/untrained) | .729/.681 | .710/.619 | .703/.613 | .701/.602 | .654/.589 | .644/.579 | .601/.580 |
| Llama (trained/untrained) | .729/.681 | .719/.590 | .682/.598 | .680/.599 | .677/.596 | .669/.561 | .654/.500 |

Table 7: Performance of a Qwen leader trained on a heterogeneous team (Qwen + Llama + Gemma), and deployed on homogeneous teams of varying sizes. Results are reported for the trained leader and untrained leader in the from trained/untrained

1. To what extent do the alternative solutions proposed by the agents contribute to the effectiveness of the trained leader?

2. Given that the agent team can generate a multitude of different responses for a single task, how many sets of such response sets should the leader be exposed to during training for each task?

3. Does filtering out easy questions improve overall training efficacy?

**Importance of Alternative Solutions**   To address the first question above, we assess the importance of alternative solutions provided by the agent team during both training and inference. As previously noted, the composition of the agent team, and thus the diversity (and quality) of solutions presented to the leader, can significantly affect performance. To investigate this, we evaluate five configurations that vary in terms of how alternative solutions are presented to the trained model (both at training and at inference time).

- *GRPO Zeroshot Data*: No alternative solutions at training or inference; model sees only its own outputs.

- *SCoRe and GRPO on Self-Reflection Data*: Model is trained and inferenced with its own solutions.

- *Zeroshot as Leader*: No alternative solutions during training; observes agent team responses at inference.

- *Our Leader w. All Qwen-2.5 Team*: Trained and inferenced with homogeneous team of Qwen-2.5 agents.

- *Our Leader w. Diverse Team*: Trained and inferenced with heterogeneous team of three distinct agents.

Table 8 presents the performance of each method after five rounds of inference. Among these, the best performance is achieved by the leader trained, via our method, with responses from a diverse team of agents. Interestingly, we observe no clear advantage when models are trained on their own solutions (as in SCoRe and GRPO Self-Reflect) compared to those trained on homogeneous agent team responses. These results highlight the importance of sourcing alterative solutions form multiple distinct agents, which results in the highest-quality responses from the leader.

**Leader's Exposure to Distinct Agent Solutions Per Task During Training**   We next examine how the number of distinct solution sets per task provided to the leader during training affects performance. We use BBH as an exemplar for this ablation. Specifically, we train the leader using 1, 4, and 8 sets of agent responses per task (in our main experiments, we adopt 4 sets per task). To ensure a fair comparison, all variants use the same total number of training examples. Table 9 reports leader accuracy under each setting. We observe that exposing the leader to multiple solution sets improves performance, though gains begin to plateau beyond 4 sets. This suggests that diverse agent responses

| Method | MMLU | BBH | MATH |
|---|---|---|---|
| Untrained | $0.734_{\pm0.006}$ | $0.733_{\pm0.010}$ | $0.666_{\pm0.002}$ |
| GRPO Zeroshot Data | $0.747_{\pm0.003}$ | $0.814_{\pm0.006}$ | $0.718_{\pm0.006}$ |
| SCoRe | $0.752_{\pm0.005}$ | $0.828_{\pm0.004}$ | $0.721_{\pm0.004}$ |
| GRPO Self-Reflect Data | $0.762_{\pm0.004}$ | $0.857_{\pm0.006}$ | $0.720_{\pm0.004}$ |
| Zeroshot as Leader | $0.742_{\pm0.003}$ | $0.783_{\pm0.002}$ | $0.729_{\pm0.007}$ |
| Our Leader w. All Qwen-2.5 Team | $0.742_{\pm0.004}$ | $0.816_{\pm0.008}$ | $\mathbf{0.767}_{\pm0.003}$ |
| Our Leader w. Diverse Team | $\mathbf{0.782}_{\pm0.006}$ | $\mathbf{0.882}_{\pm0.005}$ | $0.762_{\pm0.005}$ |

Table 8: Accuracy after 5 rounds of inference.

help the leader generalize to varied team behaviors, but that additional diversity beyond a certain point offers diminishing returns.

| Number of Solution Sets Seen at Training | | |
|---|---|---|
| 1 Set | 4 Sets | 8 Sets |
| $0.890_{\pm0.005}$ | $0.917_{\pm0.003}$ | $0.906_{\pm0.002}$ |

Table 9: Leader accuracy for different numbers of solution sets per task during training (BBH), $\pm2$ standard errors.

We also examine the impact of training-time task difficulty on leader performance. Specifically, we consider how "easy" a given task is for the agent team, independent of the leader. Any task where at least 75% of the agent responses are correct is removed from the training data. Training on these filtered tasks leads to improved leader performance at test time. One possible explanation is that including tasks on which the agent team already performs well may cause the leader to become overly reliant on agent responses, reducing its incentive to critically evaluate or override incorrect solutions.

| UNFILTERED TRAINING | FILTERED TRAINING |
|---|---|
| $0.869_{\pm0.004}$ | $0.882_{\pm0.005}$ |

Table 10: Accuracy of leader trained with and without difficulty-based filtering for BBH, with $\pm2$ standard errors.

It is worth noting these ablations cannot be directly compared to the results in the main body as a different amount of training data is used to ensue fair comparisons between each ablations.

**Solution Set Ordering** Next, we discuss the ordering in which the leader is exposed to the different sets of solutions. For our main experiments, we sample four sets of agent solutions for each task $x$. In Figure 13 we see difference in reward curves for training with shuffled solution sets, and grouped solution sets (i.e., the leader sees all four solution sets in a single batch). We find that grouping solutions by task makes training much more stable and results in higher reward for the trained leader.

**Training Configuration** For all experiments involving our leader models we use 8 H100s for RL training and between 1-4 H100s for inference and SFT. For all RL training we learning rates between 1e-6 to 5e-7, KL-regularization of $\beta = 0, 0.01, 0.05$. Each training experiment uses 1 epoch.

## C Different Choice of Leader

In the main body we use Qwen-2.5-7B-Instruct as our leader. In Tables 11 and 12, we present results when Gemma-2-9B-it and Llama-3.1-8B-Instruct are used as the leader agent, respectively.

To ensure a fair comparison against baselines when varying the leader agent in our method, we alternate the "main" model used in the baselines, e.g., for SelectLLM and the deferral leader, the

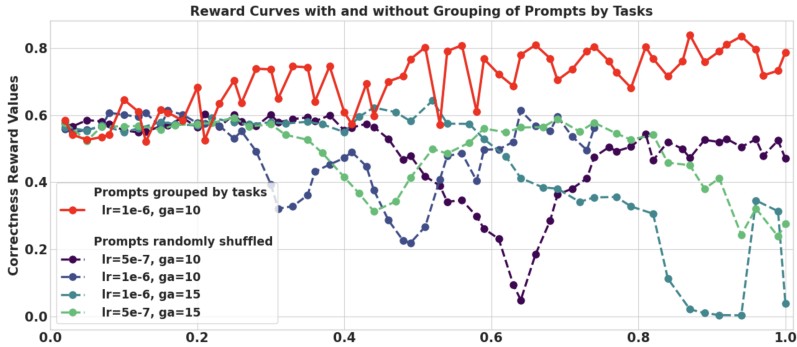

Figure 13: Reward curves for training with grouped solution sets vs randomly shuffled solution sets. Learning rate and gradient accumulation steps are denoted respectively as *lr* and *ga*.

| Type | Method | MMLU | BBH | MATH |
|---|---|---|---|---|
| Training-free | Zero-Shot | $0.748_{\pm 0.006}$ | $0.659_{\pm 0.005}$ | $0.468_{\pm 0.002}$ |
| | Self-Reflect (Madaan et al., 2023) | $0.763_{\pm 0.005}$ | $0.697_{\pm 0.005}$ | $0.481_{\pm 0.003}$ |
| | MAD (Div) (Du et al., 2023) | $\mathbf{0.771}_{\pm 0.002}$ | $0.785_{\pm 0.002}$ | $0.653_{\pm 0.009}$ |
| | MAD (Hom) (Du et al., 2023) | $\mathbf{0.776}_{\pm 0.007}$ | $0.717_{\pm 0.012}$ | $0.496_{\pm 0.008}$ |
| | Untrained Leader | $0.738_{\pm 0.004}$ | $0.759_{\pm 0.005}$ | $0.655_{\pm 0.010}$ |
| Training-based | Acc-Collab (Estornell et al., 2024a) | $\mathbf{0.769}_{\pm 0.021}$ | $0.742_{\pm 0.007}$ | $0.501_{\pm 0.006}$ |
| | SelectLLM (Maurya et al., 2024) | $0.764_{\pm 0.001}$ | $0.751_{\pm 0.001}$ | $0.549_{\pm 0.001}$ |
| | SCoRe (Kumar et al., 2024) | $0.755_{\pm 0.005}$ | $0.688_{\pm 0.007}$ | $0.452_{\pm 0.003}$ |
| | Deferal Leader | $0.741_{\pm 0.005}$ | $0.759_{\pm 0.007}$ | $0.701_{\pm 0.003}$ |
| | Zero-Shot GRPO (Zhu et al., 2024) | $0.747_{\pm 0.004}$ | $0.680_{\pm 0.003}$ | $0.440_{\pm 0.003}$ |
| | Self-Reflect GRPO (Zhu et al., 2024) + (Madaan et al., 2023) | $\mathbf{0.772}_{\pm 0.002}$ | $0.764_{\pm 0.005}$ | $0.500_{\pm 0.004}$ |
| Ours | MLPO Leader | $\mathbf{0.778}_{\pm 0.004}$ | $\mathbf{0.813}_{\pm 0.003}$ | $\mathbf{0.718}_{\pm 0.007}$ |

Table 11: Accuracy of each method on MMLU, BBH, and MATH benchmarks when Gemma-2-9B-it is used as the leader. All baselines are implemented with Gemma-2-9B-it. The leader model trained with MLPO is denoted as Our Leader.

trained model which makes the selection (deferral) is the same model our leader. For any method that uses a team (e.g., MAD) we keep the team the same.

| Type | Method | MMLU | BBH | MATH |
|---|---|---|---|---|
| Training-free | Zero-Shot | $0.703_{\pm 0.005}$ | $0.638_{\pm 0.014}$ | $0.470_{\pm 0.003}$ |
| | Self-Reflect (Madaan et al., 2023) | $0.712_{\pm 0.006}$ | $0.656_{\pm 0.014}$ | $0.484_{\pm 0.005}$ |
| | MAD (Div) (Du et al., 2023) | $\mathbf{0.771}_{\pm 0.002}$ | $0.785_{\pm 0.002}$ | $0.653_{\pm 0.009}$ |
| | MAD (Hom) (Du et al., 2023) | $0.734_{\pm 0.007}$ | $0.707_{\pm 0.015}$ | $0.533_{\pm 0.009}$ |
| | Untrained Leader | $0.714_{\pm 0.006}$ | $0.741_{\pm 0.005}$ | $0.662_{\pm 0.014}$ |
| Training-based | Acc-Collab (Estornell et al., 2024a) | $0.755_{\pm 0.020}$ | $0.740_{\pm 0.018}$ | $0.485_{\pm 0.004}$ |
| | SelectLLM (Maurya et al., 2024) | $0.766_{\pm 0.001}$ | $0.762_{\pm 0.001}$ | $0.565_{\pm 0.001}$ |
| | SCoRe (Kumar et al., 2024) | $0.524_{\pm 0.006}$ | $0.638_{\pm 0.009}$ | $0.325_{\pm 0.006}$ |
| | Deferal Leader | $0.732_{\pm 0.005}$ | $0.752_{\pm 0.007}$ | $0.692_{\pm 0.004}$ |
| | Zero-Shot GRPO (Zhu et al., 2024) | $0.714_{\pm 0.006}$ | $0.712_{\pm 0.007}$ | $0.466_{\pm 0.006}$ |
| | Self-Reflect GRPO (Zhu et al., 2024) + (Madaan et al., 2023) | $0.723_{\pm 0.006}$ | $0.721_{\pm 0.007}$ | $0.487_{\pm 0.006}$ |
| Ours | MLPO Leader | $\mathbf{0.770}_{\pm 0.006}$ | $\mathbf{0.783}_{\pm 0.009}$ | $\mathbf{0.720}_{\pm 0.007}$ |

Table 12: Accuracy of each method on MMLU, BBH, and MATH benchmarks when Llama-3.1-8B-Instruct is used as the leader. All baselines are implemented with Llama-3.1-8B-Instruct. The leader model trained with MLPO is denoted as Our Leader.

# D ADDITIONAL EXPERIMENTAL DETAILS

Here we outline our setup for training baselines.

## D.1 BASELINE DETAILS

**MAD**  Deployed with Qwen 2.5 7B Instruct, Llama 3.1 8B Instruct, and Gemma 2 9B Instruct **MAD (Div)**, and three copies of the same model (e.g., 3 copies of Qwen 2.5 7B Instruct) **MAD (Hom)**.

**Zeroshot-GRPO**  Trained with Qwen 2.5 7B Instruct for main results, and either Llama 3.1 8B Instruct, or Gemma 2 9B Instruct for Section C (selecting the same model as the leader trained with MLPO). We use a learning rate of between 1e-6 to 5e-7, KL-regularization of $\beta = 0, 0.01, 0.05$. The difficulty filtering outlined in Section B.3 is also applied to GRPO.

**ACC-Collab**  Trained two copies of Qwen 2.5 7B Instruct (one trained as the actor, one as the critic) for the main results. For results in Section C we use two copies of either Llama 3.1 8B Instruct or Gemma 2 9B Instruct (selecting the same model as the leader trained with MLPO). We train with RPO using learning rates between 1e-6 to 5e-6, a regularization term of $\alpha_{\text{RPO}} \in \{1.3, 0.7, 1\}$.

**SelectLLM**  The selection LLM is trained by taking Qwen 2.5 7B Instruct for main results (and Llama 3.1 8B Instruct, or Gemma 2 9B Instruct for Section C) and retraining the last two fully connected layers. We use a learning rate of 1e-5 with weight decay of 0.01. We deploy the variant which makes model selections via maximum confidence score, with a majority voting budget of 20 (keeping with the 20 total inference-time generations used by our pipeline for 5 rounds of inference). For results in Section 4.2.2 we increase the budget to 40 to match the number of samples used by the other methods. The selection model has access to all three agents: Qwen 2.5 7B Instruct, Llama 3.1 8B Instruct, and Gemma 2 9B Instruct.

**SCoRe**  Trained with Qwen 2.5 7B Instruct for main results, and either Llama 3.1 8B Instruct, or Gemma 2 9B Instruct for Section C (selecting the same model as the leader trained with MLPO). We use a learning rate of between 1e-6 to 5e-6, a *bonus scale* of $\alpha = 10$, stage one regularization of $\beta_1 = 0.01$, and a stage two regularization of $\beta_2 = 0.1$. In stage two we use four samples of updated answers per prompt and group these answers into the same batch (keeping with the same data grouping utilized by MLPO as outlined in Section 3.2). The difficulty filtering outlined in Section B.3 is also applied to SCoRE.

**Deferral Leader**  The leader is trained by taking Qwen 2.5 7B Instruct for main results (and Llama 3.1 8B Instruct, or Gemma 2 9B Instruct for Section C) and retraining the last two fully connected layers. The deferral leader views all solutions from the agent team and select one solution. We use a learning rate of 1e-5 with weight decay of 0.01. The deferral leader is trained on the same data that the MLPO leader is trained on (outlined in Section 3.2), and is deployed on the same team as the MLPO leader.

## D.2 COMPUTE AND TOKEN USAGE

**GPU Hours**: All training was completed using Nvidia H100 GPUs. Table 13 gives the total GPU hours required for training when using Qwen-2.5-7B-Instruct as the base model. We compute GPU hours as num-gpus*training-time:

| Dataset | ACC-Collab | SelectLLM | SCoRe | Deferral Leader | GRPO | MLPO |
|---------|-----------|-----------|-------|-----------------|------|------|
| MMLU | 34.2 | 18.1 | 218.3 | 17.6 | 112.3 | 172.2 |
| BBH | 21.1 | 17.2 | 127.2 | 11.1 | 82.6 | 100.5 |
| MATH | 25.7 | 15.9 | 189.9 | 18.3 | 120.5 | 168.7 |

Table 13: Total GPU training hours (computed as num-gpus*training-time) for each method.

**Inference-Time Token Usage**: Table 14 shows the tokens used by each method

Table 14: Token usage for answering one question after $r=0$ and $r=4$ rounds (averaged across all examples for each dataset). SelectLLM is computed with a budget of 20

| Method | MATH ($r=0$ / $r=4$) | BBH ($r=0$ / $r=4$) | MMLU ($r=0$ / $r=4$) |
|---|---|---|---|
| Self-reflect GRPO | 2.5k / 12k | 2k / 9k | 1.5k / 8k |
| Div MAD | 6k / 20k | 4k / 14k | 3k / 16k |
| SelectLLM | – / 18k | – / 17k | – / 19k |
| ACC-Collab | 8k / 26k | 5k / 18k | 4k / 21k |
| Untrained leader | 9k / 38k | 7k / 30k | 6k / 27k |
| SCoRE | 4k / 20k | 3k / 12k | 3k / 14k |
| MLPO Leader (ours) | 12k / 50k | 10k / 40k | 9k / 34k |

### D.3   SFT AND BACKTRACKING

**MLPO:** Details on SFT dataset construction and prompts can be found in Section E.3. For MLPO we utilize an SFT dataset to teach the model backtracking.

**Baselines:** For the GRPO baseline we also follow an identical procedure to produce an SFT backtracking dataset. The baselines SelectLLM and Deferral Leader also make use of this, but only the prompts (not completions), since they are classifiers. ACC-Collab does not use an SFT set as the training scheme uses a mix of DPO and SFT durring training. SCoRE has its own two-phase training pipeline which we follow. Table 15 the size, and average number of tokens, used in the SFT pretraining datasets for MLP and GRPO

| Dataset | Examples (GRPO/MLPO) | GRPO Prompt | MLPO Prompt | GRPO Response | MLPO Response |
|---|---|---|---|---|---|
| MMLU | 7,431 / 7,470 | 185.7 | 976.3 | 1650.4 | 1846.3 |
| BBH | 2,211 / 2,192 | 189.7 | 1072.5 | 1430.6 | 1665.5 |
| MATH | 4,603 / 4,744 | 204.3 | 1301.3 | 1488.8 | 1552.5 |

Table 15: Number of examples and average tokens in prompt and response for the SFT data used for MLPO and GRPO

## E   PROMPTS USED IN OUR PIPELINE

In this section, we present the prompts used in our pipeline, with examples drawn specifically from the MATH dataset. Analogous prompt structures are used for the other datasets.

### E.1   AGENT PROMPTS

We use structured and context-aware prompts for the agent models, both during inference and while collecting the training data. The prompts differ slightly depending on the inference round.

**Round 0 Prompt**   In the first round, agents receive the question along with a standard solving plan. The prompt encourages step-by-step reasoning, adherence to factual premises, and concludes with a boxed answer.

---

**Agent Prompt for Round 0**

```
You are part of a team of 3 LLMs responsible for solving a
math problem.  You are Agent_[agent_number].  Strictly follow
the given plan to solve the following math problem by thinking
step by step through the plan.  Make sure to first define the
premises of the question, and make sure that the answer is
consistent with the premises.  Strictly stick to the facts and
```

```
question provided.

*** Question:  [question text] ***

*** Plan:  Solve the given math problem.  Think step-by-step,
providing detailed calculations and reasoning for your steps.
***

Regardless of the approach, always conclude with:

Therefore, the final answer is:  $\boxed{[answer]}$.  Where
[answer] is just the final number or expression that solves the
problem.
```

**Subsequent Round Prompts**  In later rounds, agents are additionally provided with their previous solution and feedback from the aggregator (leader). This guides the agent to reflect, revise, and correct any issues based on the leader's evaluation and questions.

---

**Agent Prompt for Subsequent Rounds**

```
You are part of a team of 3 LLMs collaborating to solve a
math problem.  You are Agent_[agent_number].  Your goal
is to improve your previous response using feedback in
the form of questions.  Strictly follow the plan and think
step-by-step through the math problem.  Make sure your answer
is consistent with the premises and addresses the raised
questions thoroughly.

*** Question:  [question text] ***

Your previous solution:  [previous_solution]

Additionally, the aggregator has evaluated agent responses from
the previous round and has also raised questions about your
previous response.

The aggregator's output:  [aggregator_response]

*** Plan:

1.  Carefully reflect on the aggregator's feedback and your
previous solution.

2.  Revise your answer step-by-step to improve its correctness
and clarity.  Address each question raised where relevant.

3.  Double-check for any logical, calculation, or reasoning
errors.

Regardless of the approach, always conclude with:

Therefore, the final answer is:  $\boxed{[answer]}$.  Where
[answer] is just the final number or expression that solves the
problem.
```

## E.2 LEADER PROMPTS

The leader model is prompted to evaluate the responses of all agents to the given problem and produce a final answer. The prompt instructs the leader to critically analyze the agents' outputs, identify correct reasoning or mistakes, and then synthesize a coherent and accurate solution. The leader is explicitly instructed to produce two structured blocks: a reasoning trace enclosed in `<think>` tags, and a final answer in the `<answer>` block, which must conclude with a boxed expression.

---

**Leader Prompt**

```
You are an expert aggregator LLM tasked with evaluating
multiple agents' responses to a math problem.  Your goal is
to critically analyze all agent responses, identify correct
reasoning or errors, and then provide a unified answer.

Question:  [question text]

Agent 1 Response:  [agent_1 response]

Agent 2 Response:  [agent_2 response]

Agent 3 Response:  [agent_3 response]

Please complete the following two blocks in order:

1.  <think>...</think>:  A long, detailed chain-of-thought
reasoning process.

2.  <answer>...</answer>:  Your final answer should be
aggregated from the best elements of the agents' responses.

- End the answer with:  Therefore, the final answer is:
$\boxed{[answer]}$.
```

---

## E.3 GENERATING BACKTRACKING DATA FOR SFT

To teach the leader model natural backtracking and self-correction behavior, we construct a supervised fine-tuning (SFT) dataset comprising synthetic completions that mimic realistic reasoning failures followed by recovery. This is inspired by recent work encouraging naturalistic self-correction patterns in LLMs Qin et al. (2024); Guo et al. (2025); Team et al. (2025).

We begin by running our untrained multi-agent pipeline to collect 16 distinct completions from the leader for each agent team generation. We discard examples where all completions are incorrect or all correct. For the remaining examples, we randomly select one correct and one incorrect leader completion. These are then used to construct a prompt that asks an untrained leader model to imitate a reasoning process that starts with plausible but flawed logic (using the incorrect trajectory), then backtracks and self-corrects to reach the correct answer (using the correct trajectory). We then filter out any remaining examples where the backtracking did not yield a correct final answer. The generated backtracked completions form the core of our SFT dataset.

Below, we include the exact prompt used to generate such backtracking examples for the MATH dataset.

---

**Backtracking Generation Prompt (for MATH)**

```
You are an AI aggregator tasked with evaluating multiple agent
responses to a question and aggregating them into a coherent
```

---

and accurate final answer.  You will be provided with the original prompt of the aggregator, a correct aggregation (which evaluated the agents correctly and arrived at the correct final answer), and an incorrect aggregation (which arrived at an incorrect final answer).  Your goal is to mimic an aggregator which tries to aggregate the agent responses, first makes mistakes in its reasoning process by going down the wrong path and then backtracks to the correct reasoning.

The task consists of the following components:
1.  **Prompt**:  The original math question and agent responses.
2.  **Incorrect aggregator response**:  Response from a previous aggregator that contains errors in aggregation or reasoning and hence arrives at an incorrect final answer. These should be carefully reviewed for mistakes.
3.  **Correct aggregator response**:  Response from a previous aggregator that contains the correct reasoning, evaluation and aggregation.

Here's the input data for your aggregation:
Question:  [question text]
Agent 1 response:  [agent_1 response]
Agent 2 response:  [agent_2 response]
Agent 3 response:  [agent_3 response]
Incorrect reasoning from Previous Aggregator:  [extracted <think> block from incorrect leader response]
Correct reasoning from Previous Aggregator:  [extracted <think> block from correct leader response]

Here's your task breakdown:
- You should start by examining the agent responses and considering the provided aggregator responses.
- **IMPORTANT**:  You should use the incorrect aggregator response to mimic an aggregator which makes mistakes in its reasoning process before backtracking in your reasoning, acknowledging the mistakes, and correcting your approach based on the correct aggregator response provided.
   - As you work through the problem, narrate your thinking naturally just like a student working things out on paper. When you notice a mistake, acknowledge it with a natural reflection like:
      - "Wait, that doesn't seem right."
      - "Hmm, I think I made a mistake there."
      - "Actually, let me go back and check that."
      - "Maybe I should rethink this."
Your solution should feel like a continuous, thoughtful monologue that includes:
- One or more initial (realistic) mistakes in reasoning.
- Natural moments of self-correction and backtracking.
- Accurate evaluation of agent correctness (as provided in the correct response).
- A correct final solution (as provided in the correct response).

At the end of your response you should include the final answer as:  Final answer:  $\boxed{[answer]}$

```
**IMPORTANT**:  Your final answer must be consistent with the
correct aggregator's final answer.
```

**Example Backtracking Generation**  Below, we include an example used in our SFT dataset generation.  The first box shows the prompt, which includes the agent team's responses for a given MATH question, and the second box contains the backtracked response generated using the methodology outlined above, which is used as the completion to SFT the leader for the given prompt below.

---

**Example Prompt with Task + Agent Team Generation**

```
You are an expert aggregator LLM tasked with evaluating
multiple agents' responses to a math problem.  Your goal is
to critically analyze all agent responses, identify correct
reasoning or errors, and then provide a unified answer.

Question:  Alex needs to catch a train.  The train arrives
randomly some time between 1:00 and 2:00, waits for 10 minutes,
and then leaves.  If Alex also arrives randomly between 1:00
and 2:00, what is the probability that the train will be there
when Alex arrives?

Agent 1 Response:  **Problem Definition**
To solve this problem, we need to understand the conditions for
Alex to catch the train.  The train arrives randomly between
1:00 and 2:00, waits for 10 minutes, and then leaves.  Alex
also arrives randomly between 1:00 and 2:00.

**Step 1:  Define the Time Interval**
Let's define the time interval from 1:00 to 2:00 as a 60-minute
window.  We can represent this interval as a continuous segment
from 0 to 60 minutes, where 0 represents 1:00 and 60 represents
2:00.

**Step 2:  Calculate the Total Time Interval**
The total time interval for both the train's arrival and Alex's
arrival is 60 minutes.

**Step 3:  Calculate the Train's Waiting Time**
The train waits for 10 minutes after its arrival.  This means
that if the train arrives at time 't', it will be present at
the station from 't' to 't + 10' minutes.

**Step 4:  Define the Conditions for Alex to Catch the Train**
For Alex to catch the train, he must arrive during the
10-minute waiting period after the train's arrival.  This means
that Alex's arrival time 'A' must satisfy the condition:
1 ≤ A ≤ t + 10, where t is the train's arrival time.

**Step 5:  Calculate the Probability**
To calculate the probability, we need to find the area of the
region where Alex arrives while the train is present.  This can
be visualized as a shaded area in a 2D coordinate system, where
the x-axis represents the train's arrival time and the y-axis
represents Alex's arrival time.
The total area of the sample space is a square with side length
60, representing the total time interval.
```

The region where Alex catches the train is a trapezoid with bases 60 and 50 and height 10.

**Step 6: Calculate the Area of the Trapezoid**
The area of a trapezoid is given by:
Area = (1/2) × (base1 + base2) × height = (1/2) × (60 + 50) × 10 = (1/2) × 110 × 10 = 550

**Step 7: Calculate the Total Area of the Sample Space**
The total area of the sample space is a square with side length 60, so the area is:
Total Area = 60 × 60 = 3600

**Step 8: Calculate the Probability**
The probability of Alex catching the train is the ratio of the area of the trapezoid to the total area of the sample space:
Probability = Area of Trapezoid / Total Area = 550 / 3600 = 55/360 = 11/72

**Conclusion**

Therefore, the final answer is: $\boxed{\dfrac{11}{72}}$.

**Agent 2 Response**: Here's a step-by-step solution to the problem:
**1. Define the Sample Space:**
The sample space consists of all possible pairs of arrival times for the train and Alex. Since both can arrive anytime between 1:00 and 2:00, we have a range of 60 minutes for each.

**2. Visualize the Problem:**
Imagine a square where the x-axis represents the train's arrival time and the y-axis represents Alex's arrival time. Each point within the square represents a possible combination of arrival times.

**3. Identify Favorable Outcomes:**
We want the cases where Alex arrives within the 10-minute window when the train is waiting. This forms a 10-minute wide strip along the line where the train's arrival time is equal to Alex's arrival time minus 10 minutes.

**4. Calculate Areas:**
* **Total Area:** The total area of the square represents all possible outcomes, which is 60 minutes * 60 minutes = 3600 square minutes.
* **Favorable Area:** The favorable area is a 60-minute long strip, with a width of 10 minutes. So, the favorable area is 60 minutes * 10 minutes = 600 square minutes.

**5. Calculate Probability:**
Probability is the ratio of favorable outcomes to total outcomes:
Probability = (Favorable Area) / (Total Area) = 600 square minutes / 3600 square minutes = 1/6

Therefore, the final answer is: $\boxed{\dfrac{1}{6}}$.

**Agent 3 Response:** To solve this problem, we need to determine the probability that Alex will arrive at the train station during the time the train is present. We can visualize this problem using a coordinate plane where both the train's arrival time and Alex's arrival time are plotted.

### Step 1: Define the time intervals
– The train arrives at a random time between 1:00 and 2:00. We can represent this time interval as $[0, 60]$ minutes past 1:00. – Similarly, Alex arrives at a random time between 1:00 and 2:00, which we can also represent as $[0, 60]$ minutes past 1:00.

### Step 2: Set up the coordinate system
We can plot the arrival times of the train and Alex on a coordinate plane where the x-axis represents the train's arrival time and the y-axis represents Alex's arrival time. Both axes range from 0 to 60 minutes.

### Step 3: Define the region of interest
The train waits for 10 minutes after its arrival, so the train will be at the station from $t$ to $t + 10$, where $t$ is the train's arrival time. Alex will catch the train if he arrives during this 10-minute window.

### Step 4: Identify the favorable region
We need to find the area on the coordinate plane where Alex arrives during the train's 10-minute window. This area can be visualized as a band that runs horizontally from $t$ to $t + 10$ for each $t$ from 0 to 50 (since if the train arrives at 50 minutes past 1:00, it will leave at 60 minutes past 1:00, and Alex can still catch it if he arrives at 60 minutes).

### Step 5: Calculate the area of the favorable region
The favorable region is a band that is 10 units wide and 50 units long (from 0 to 50 on the x-axis). The area of this region is:

$$\text{Area of favorable region} = 10 \times 50 = 500 \text{ square units}$$

### Step 6: Calculate the total possible region
The total possible region is a square with side length 60, so the area is:

$$\text{Area of total region} = 60 \times 60 = 3600 \text{ square units}$$

### Step 7: Determine the probability
The probability that Alex will catch the train is the ratio of the area of the favorable region to the area of the total region:

$$\text{Probability} = \frac{\text{Area of favorable region}}{\text{Area of total region}} = \frac{500}{3600} = \frac{5}{36}$$

```
### Conclusion

Therefore, the final answer is:  $\boxed{\dfrac{5}{36}}$ .

Please complete the following three blocks in order:
1.  <think>...</think>:  A long, detailed chain-of-thought
reasoning process.
2.  <answer>...</answer>:  Your final answer should be
aggregated from the best elements of the agents' responses.
- End the answer with:  'Therefore, the final answer is:
$\boxed{[answer]}$'
Your output should follow this format:
<think>
[detailed thought process of the aggregator]
</think>
<answer>
[The final unified aggregated response with a concise
justification.  Conclude with:  'Therefore, the final answer
is:  $\boxed{[answer]}$'.]
</answer>
```

**Leader's Completion with Backtracking**

```
Let's approach this problem step-by-step, and I'll narrate my
thinking along the way.  I'll start by examining the agent
responses and identifying any mistakes I might have made.

### Step 1:  Define the Sample Space
Both the train and Alex can arrive at any time between 1:00
and 2:00.  Represent this as a 60-minute interval on both axes.
This gives us a square with side length 60, representing all
possible pairs of arrival times.

### Step 2:  Identify the Favorable Outcomes
The train waits for 10 minutes after its arrival, so Alex can
catch the train if he arrives between $t$ and $t + 10$, where $t$ is
the train's arrival time.  This forms a band on the coordinate
plane where Alex's arrival time $y$ is within 10 minutes of the
train's arrival time $x$.

### Step 3:  Visualize the Problem
We can plot this on a coordinate plane.  The region of interest
is a band that runs horizontally from $t$ to $t + 10$ for each $t$ from
0 to 50.  This is because if the train arrives at 50 minutes
past 1:00, it will leave at 60 minutes past 1:00, and Alex can
still catch it if he arrives at 60 minutes.

### Step 4:  Calculate the Areas
- **Total Area:** The total area of the square represents all
possible outcomes, which is $60 \times 60 = 3600$ square minutes.
- **Favorable Area:** The favorable area is a band that is 10
units wide and 50 units long (from 0 to 50 on the x-axis).  The
area of this region is $10 \times 50 = 500$ square units.

### Step 5:  Determine the Probability
```

The probability is the ratio of the area of the favorable region to the area of the total region:

$$\text{Probability} = \frac{\text{Area of favorable region}}{\text{Area of total region}} = \frac{500}{3600} = \frac{5}{36}$$

### Reflection and Correction
Wait, that doesn't seem right. Agent 1 and Agent 3 both calculated the favorable area as 550 and 500 square units, respectively. I should double-check my calculation.

### Detailed Recheck
Let's re-examine the region of interest. The train arrives between 0 and 60 minutes, and for each $t$, Alex can arrive between $t$ and $t + 10$. This forms a trapezoid with vertices at (0,0), (10,0), (60,50), and (50,60).
- The area of the trapezoid can be calculated as:

$$\text{Area} = \frac{1}{2} \times (\text{base1} + \text{base2}) \times \text{height}$$

Here, base1 = 60 (from (0,0) to (60,0)), base2 = 50 (from (10,0) to (60,50)), and height = 10.

$$\text{Area} = \frac{1}{2} \times (60 + 50) \times 10 = \frac{1}{2} \times 110 \times 10 = 550$$

### Final Calculation
The total area is still 3600 square units. Therefore, the probability is:
$$\text{Probability} = \frac{550}{3600} = \frac{11}{72}$$

### Final Answer
After rechecking and correcting my initial mistake, I now see that the correct answer is indeed $\frac{11}{72}$.

Final answer: $\boxed{\frac{11}{72}}$

</think>
<answer>
Based on the detailed analysis, only Agent 1 provided the correct reasoning and final answer. Therefore, the final answer is: $\boxed{\frac{11}{72}}$.
</answer>

