# OpenReview forum: "How to Train a Leader: Hierarchical Reasoning in Multi-Agent LLMs"
_ICLR.cc/2026/Conference — Submitted to ICLR 2026_

### Official Review · Reviewer_D7xX · 2025-10-15

**Soundness:** 3
**Presentation:** 3
**Contribution:** 2
**Rating:** 6
**Confidence:** 2

**Summary:**

This paper proposed a novel fine-tuning scheme for llm agent, in which only a leader model is trained while the agent team maintains their own parameters. The overall scheme is essentially fine-tuning an answer aggregator for downstream tasks. The framework is validated in three tasks, including BBH, MATH, and MMLU.

**Strengths:**

1. The motivation of this paper, which focuses on improving the computational burden of a multi-LLM-agent system, is meaningful.
2. The fine-tuning scheme towards multi-LLM-agent collaboration is elegant and novel
3. The paper is well-organized to present the contributions.
4. Extensive experiments validate the efficiency of the proposed framework.

**Weaknesses:**

1. The trade-off of the size of the agent team should be discussed.
2. The idea of only fine-tuning a single model in mult-llm-agent is based on empirical observation without a theoretical foundation.
3. It is not certain whether the proposed scheme is also efficient for larger llms, e.g., 32B.

**Questions:**

1. According to Figure 2, the multi-agent team is not generating new data during MLPO. Why is this setting advocated?

---

> ### Author Response · Authors · 2025-11-21
> **Author Rebuttal (1)**
>
> - _The trade-off of the size of the agent team should be discussed._
>
> **Response [Team size]** We agree that team size is an important consideration. We have provided updated results in the draft (Appendix B.2, Line 1178, Line 1223, Table 5, and Table 6) and also reported them here. We note that in the initial draft, we do study different team _types_ although an analysis of team _size_ was omitted.
>
> For the experiments below, we pair our trained leader and an untrained leader, with teams of varying size. To isolate the effects of team size, rather than team composition, we stick to homogeneous teams of $K$ agents for $0\leq K \leq 6$. We observe that
> 1. Leader performance is quite robust to the number of agents, although we do see a decrease as $K$ becomes large (5 or 6). This is likely due to either an overload of the leader's context, or an innate susceptibility  of LLMs to agree with majority opinions as more opinions are presented (as outlined in these papers  [[1](https://arxiv.org/pdf/2410.12428), [2](https://openreview.net/pdf?id=sy7eSEXdPC)])
> 2. For some weaker teams (e.g., Llama on MATH) it is better to deploy the leader as a solo agent (or with a diverse team as outlined in our main text). This lends more support to our hypothesis that diverse teams are generally the most effective.
>
>
> #### MMLU
> | Agent Team | 0 (No Team) | 1 | 2 | 3 | 4 | 5 | 6 |
> | :- | :-: | :-: | :-: | :-: | :-: | :-: | :-: |
> | Qwen-2.5-7B-Instruct (trained/untrained)       | 0.757/0.746 | 0.757/0.698 | 0.758/0.723 | 0.760/0.720 | 0.761/0.722 | 0.752/0.699 | 0.748/0.681 |
> | Gemma-2-9b-it (trained/untrained)              | 0.757/0.746 | 0.760/0.741 | 0.762/0.751 | 0.762/0.757 | 0.759/0.756 | 0.757/0.750 | 0.750/0.743 |
> | Meta-Llama-3.1-8B-Instruct (trained/untrained) | 0.757/0.746 | 0.752/0.729 | 0.750/0.730 | 0.745/0.738 | 0.743/0.727 | 0.741/0.732 | 0.737/0.719 |
>
>
> #### BBH
> | Agent Team | 0 (No Team) | 1 | 2 | 3 | 4 | 5 | 6 |
> | :- | :-: | :-: | :-: | :-: | :-: | :-: | :-: |
> | Qwen-2.5-7B-Instruct   (trained/untrained)     | 0.855/0.766 | 0.856/0.753 | 0.856/0.750 | 0.858/0.759 | 0.856/0.760 | 0.853/0.763 | 0.849/0.747 |
> | Gemma-2-9b-it         (trained/untrained)      | 0.855/0.766 | 0.858/0.751 | 0.866/0.751 | 0.868/0.750 | 0.865/0.750 | 0.859/0.755 | 0.843/0.741 |
> | Meta-Llama-3.1-8B-Instruct (trained/untrained) | 0.855/0.766 | 0.860/0.755 | 0.860/0.751 | 0.862/0.755 | 0.857/0.756  | 0.840/0.765 | 0.839/0.763 |
>
>
> #### MATH
> | Agent Team | 0 (No Team) | 1 | 2 | 3 | 4 | 5 | 6 |
> | :- | :-: | :-: | :-: | :-: | :-: | :-: | :-: |
> | Qwen-2.5-7B-Instruct       (trained/untrained) | 0.729/0.681 | 0.758/0.755 | 0.756/0.751 | 0.761/0.759 | 0.755/0.754 | 0.758/0.762 | 0.756/0.754  |
> | Gemma-2-9b-it              (trained/untrained) | 0.729/0.681 | 0.710/0.619 | 0.703/0.613 | 0.701/0.602 | 0.654/0.589 | 0.644/0.579 |  0.601/0.580 |
> | Meta-Llama-3.1-8B-Instruct (trained/untrained) | 0.729/0.681 | 0.719/0.590 | 0.682/0.598 | 0.680/0.599 | 0.677/0.596 | 0.669/0.561 | 0.654/0.50 |
>
>
>
>
> - _The idea of only fine-tuning a single model in mult-llm-agent is based on empirical observation without a theoretical foundation._
>
> **Response [Theoretical Foundation]** We agree that it would be nice to have a stronger theoretical grounding for MLPO. However, we note that our goal is not to produce a theory of _why_ only a single model should be trained. Rather, our goal is to provide an effective framework for training only a single (leader) agent. Our experiments demonstrate that we can indeed achieve strong results while only training this one agent.
>
> While we do not aim to provide theoretical results or justification for this design choice, we note that from a conceptual perspective, the use of a hierarchical structure puts the leader in a central role where it controls the flow of information from round to round, thus giving it higher impact on the efficacy of this joint leader-team pipeline. Due to this central role, we thus expect training the leader to have a greater effect on overall efficacy than training one of the agents in the agent team (or training one agent in a “flat” inference pipeline like that of MAD).
>
>
> - _It is not certain whether the proposed scheme is also efficient for larger llms, e.g., 32B._
>
> **Response [Model Size]** We agree that it remains to be seen whether MLPO is effective for larger models. Given the scope of this project, some omissions are necessary. However, we do observe consistent improvement across different architectures (Qwen, Llama, Gemma) across different datasets (MATH, MMLU, BBH) as well as different teams (hom, and het).

---

> > ### Author Response · Authors · 2025-11-21
> > **Author Rebuttal (2)**
> >
> > - _According to Figure 2, the multi-agent team is not generating new data during MLPO. Why is this setting advocated?_
> >
> > **Response [Agent Responses During Training]** This is a great question and strikes at one of our key motivations, namely the practicality of training. Requiring 4 models to be held in memory while performing an already expensive training (RL-based training), would significantly increase the overhead of our method. We find that pre-computing agent responses, rather than computing them online, still results in quite high performance (as shown by our results). Additionally, we also provide a variant of MLPO that uses an additional step where agent responses are recomputed after the leader provides a response (MLPO+). We find that this additional step can result in better performance (Table 2), but the gains are quite minimal (1%-4%). Thus, lending more strength to the idea that one needs only use round 0 responses during training, which is already sufficient to beat consistently beat or tie with all baselines across each of the datasets and models we study (Tables 1, 7, 8).

---

> > > ### Comment · Reviewer_D7xX · 2025-11-21
> > > **Response to retbuttal (2)**
> > >
> > > Thank you for your response to my question. Based on your explanation, the additional response recomputation step provides a gain of roughly 1–4%, which you consider not significant. However, according to Table 1 in the paper, the improvement of the proposed model over the baseline is only up to about 7%. Given this, it seems important to acknowledge that the online data does contribute meaningful improvement, or otherwise recognize that the overall effectiveness of the proposed model may be limited.

---

> > ### Comment · Reviewer_D7xX · 2025-11-21
> > **Response to rebuttal (1)**
> >
> > Thank you for the response. Some questions remain as follows:
> > **Team size** It seems that there is no need to be a team, given the performance.
> >
> > **Model size**  My concern regarding model size is whether a larger model could answer these questions without requiring a team-based framework. If your approach demonstrates strong performance for a model of this scale, it would be helpful to discuss how it compares with much larger models such as GPT-4o or Gemini. I suggest emphasizing either the practical value of your method in reducing computational cost relative to these larger models, or its potential to generalize and scale effectively when applied to larger architectures.

---

> > > ### Author Response · Authors · 2025-12-03
> > > **Response to Reviwer**
> > >
> > > - _ ... Based on your explanation, the additional response recomputation step provides a gain of roughly 1–4%, which you consider not significant. However, according to Table 1 in the paper, the improvement of the proposed model over the baseline is only up to about 7%. Given this, it seems important to acknowledge that the online data does contribute meaningful improvement, or otherwise recognize that the overall effectiveness of the proposed model may be limited._
> > >
> > > **Response [MLPO and MLPO+]**
> > > For ease of discussion, we have copied and pasted the untrained, MLPO, and MLPO+ results below.
> > >
> > > | | MMLU | BBH | MATH|
> > > |--|--|--|--|
> > > |Untrained Leader | 0.731 | 0.764 | 0.697|
> > > |MLPO Leader | 0.782 | 0.882 | 0.762 |
> > >  |MLPO+ Leader | 0.792 | 0.920 | 0.771 |
> > >
> > > We believe that the improvements from MLPO+ are significant, which is precisely why we included them in our main body. However, the point we are trying to make is that most of the improvement can be gained by training on round 0 prompts alone without any need to recompute agent responses (i.e., MLPO). When looking at the improvement over the untrained leader, MLPO has roughly 7.0% improvement while MLPO+ has 8.3%. Similarly, for BBH MLPO has 15.4% and MLPO+ has 20.42%; for MATH MLPO has 9.3% and MLPO+ has 10.6%. Thus, one can generally improve the performance by utilizing MLPO+, but the bulk of the improvement can be gained just by MLPO.
> > > Since both MLPO and MLPO+ are our contributions, we see the additional gain from MLPO+ as a strength of our method as users can choose which version to deploy based on their use case and available compute.
> > >
> > >
> > >
> > >
> > >
> > > - _Team size It seems that there is no need to be a team, given the performance._
> > >
> > > **Response [Team Size and Need for the Team]** While the leader can achieve high performance even without the team, the highest performance is generally attained when the leader is deployed with the team. Moreover, the above results on team size are for homogeneous teams, which we observe to perform generally worse than a heterogeneous team of size 3, even as the size of a homogeneous team is scaled up (hence our focus on a heterogeneous team in the main paper).
> > >
> > > When examining the zero-shot performance of the leader (i.e., leader with no team), we agree that the leader performs quite well. We see this as a key strength of MLPO, when inference time compute is more limited, one can deploy the leader as a solo agent without multiple rounds of inference and attain good performance. If computers is more abundant, then one can deploy the leader with the team and with multi-round inference (which generally yields the highest results). This flexibility is one of the key strengths of our method!
> > >
> > >
> > > - _Model size My concern regarding model size is whether a larger model could answer these questions without requiring a team-based framework. If your approach demonstrates strong performance for a model of this scale, it would be helpful to discuss how it compares with much larger models such as GPT-4o or Gemini. I suggest emphasizing either the practical value of your method in reducing computational cost relative to these larger models, or its potential to generalize and scale effectively when applied to larger architectures._
> > >
> > >
> > > **Response [Larger Models vs MLPO]** We have conducted experiments comparing our trained Qwen leader with a heterogeneous team (MLPO) vs the performance of some larger single models; with this setup, our method uses 7B + 8B + 9B + 7B = 31B parameters.
> > >
> > > We find that in many cases MLPO attains superior performance compared to these larger models, although in some cases MLPO has worse performance. On the more reasoning-heavy benchmarks (MATH and BBH) MLPO beats these larger models in 7 out of 8 case, while on the more fact-based benchmark (MMLU) MLPO loses to the large models in 3 out of 4 cases on MMLU. These results demonstrate that MLPO is capable of beating larger models (specifically in reasoning-heavy domains), and give more evidence to the claim that MLPO teaches the model more than mere factual memorization or selection of the most “correct” agent.
> > >
> > > Below we indicate the number of larger models which MLPO beats with “(num-models-beated/total-number-of-models)”.
> > >
> > > | Model/Method | MMLU | BBH  | MATH
> > > |:--:|:--:|:--:|:--:|
> > > |MLPO              | 0.782 (1/4)  | 0.882 (4/4)  | 0.762 (3/4)
> > > |Gemma-2-27B | 0.775   | 0.725  | 0.514
> > > |Llama-3-70B    | 0.860   | 0.848  | 0.731
> > > |DeepSeek-70B| 0.840   | 0.866  | 0.771
> > > |GPT-4o            | 0.861   | 0.826  | 0.754

---

### Official Review · Reviewer_Sy98 · 2025-10-31

**Soundness:** 3
**Presentation:** 3
**Contribution:** 2
**Rating:** 4
**Confidence:** 3

**Summary:**

The paper presents a hierarchical setup where one main “leader” model coordinates a small group of helper agents. Only the leader is trained, while the others stay fixed. The leader first goes through supervised fine-tuning using synthetic data, then is further improved with a GRPO-like reinforcement objective (called MLPO). During testing, the leader communicates with the helper agents for T rounds to combine their responses into a final answer. On the task BBH, MATH, and MMLU, the approach reports gains over training‑free and training‑based baselines, and intriguingly the trained leader also improves in zero‑shot (solo) mode (Table 1; Fig. 4).

**Strengths:**

1. Train one coordinator that reads multiple agent drafts and outputs a consolidated answer; the inference workflow is well depicted (Fig. 1 on p. 3; Fig. 2 on p. 4).
2. On all three benchmarks the leader trained with SFT+MLPO outperforms strong single‑ and multi‑agent baselines (Table 1 on p. 6).
3. Analyses of team diversity (heterogeneous vs. homogeneous), exposure to alternative solutions, and information sharing (reasoning vs. final answers) provide insight (Appx B.2–B.3; Fig. 11–12 on pp. 21–22).

**Weaknesses:**

1. The evaluation scope is relatively narrow, which lacks evidence of generality. Experiments cover only BBH, MMLU, MATH with 7–9B models; there is no evidence for code, retrieval/tool use, long‑context, or multilingual settings. Claims about “collaborative reasoning” need diverse tasks; current evidence mainly supports multiple‑choice and math settings in English. Other works on the same topic, such as Mixture-of-Agents [1] and ACC-Collab [2], usually cover a wider range of task types or provide more detailed comparative analyses to better support their claims about the generality of “collaborative reasoning.”

2. Although the paper trains only a single “leader,” its inference protocol requires a team of K=3 agents and T=5 sequential interaction rounds (leader–agents–leader …), which implies ~20 generations per query. The main text fixes T=5 (Sec. 3.1) and compares methods under a ceiling of “<= 40 generations with majority vote” (Fig. 3; Appx. Fig. 7), but it does not report token‑normalized or time‑normalized efficiency (e.g., wall‑clock latency, total prompt+completion tokens, or throughput) for the proposed method versus baselines. Appendix B.3 lists training hardware/hyper‑parameters but likewise omits inference latency/throughput; the conclusion explicitly acknowledges increased context lengths, higher inference compute, and reduced parallelizability due to sequential leader–agent interactions. Together, these omissions prevent a rigorous cost–quality comparison.

**Questions:**

I suggest that the authors can report accuracy vs. tokens/latency for varying T, K, and leader/agent sampling to characterize real‑world cost trade‑offs; include token‑normalized comparisons for baselines (extend Fig. 3/7). Moreover, they can document baseline tuning budgets; where margins are small (e.g., MATH), report sensitivity to temperature/top‑p, number of rounds, and aggregation schemes; include paired‑seed significance tests (Table 1).

---

> ### Author Response · Authors · 2025-11-21
> **Author Rebuttal**
>
> - _The evaluation scope is relatively narrow, which lacks evidence of generality. Experiments cover only BBH, MMLU, MATH with 7–9B models; there is no evidence for code, retrieval/tool use, long‑context, or multilingual settings. Claims about “collaborative reasoning” need diverse tasks; current evidence mainly supports multiple‑choice and math settings in English. Other works on the same topic, such as Mixture-of-Agents [1] and ACC-Collab [2], usually cover a wider range of task types or provide more detailed comparative analyses to better support their claims about the generality of “collaborative reasoning.”_
>
> **Response [Evaluation Scope]** We agree that adding additional benchmarks and model sizes would broaden the scope of work; however, we do study 3 benchmarks which cover a wide range of topics (BBH and MMLU are constructed from a mixture of many domains, and MATH covers open-ended mathematical reasoning). We chose to focus the breadth of our experiments on different leaders (Qwen, Llama, Gemma), team compositions (heterogeneous and homogeneous teams), training ablation (difficulty filtering, solution grouping, multi-round training, number of solutions per question), and generalization of the leader (improved zero-shot capabilities).
>
> Lastly we note that while a work like ACC-Collab [2] may use _more_ benchmarks, the coverage of these benchmarks is not broader than the three used in our paper. [2] makes use of BoolQ, MMLU, BBH, SCIQ, and ARC; all of which are multiple choice QA benchmarks. Moreover the questions covered by ARC and SCIQ (general science benchmarks) are similar in nature to those covered in MMLU. They lack a dedicated mathematical benchmark like MATH.
>
> - _Although the paper trains only a single “leader,” its inference protocol requires a team of K=3 agents and T=5 sequential interaction rounds (leader–agents–leader …), which implies ~20 generations per query. The main text fixes T=5 (Sec. 3.1) and compares methods under a ceiling of “<= 40 generations with majority vote” (Fig. 3; Appx. Fig. 7), but it does not report token‑normalized or time‑normalized efficiency (e.g., wall‑clock latency, total prompt+completion tokens, or throughput) for the proposed method versus baselines. Appendix B.3 lists training hardware/hyper‑parameters but likewise omits inference latency/throughput; the conclusion explicitly acknowledges increased context lengths, higher inference compute, and reduced parallelizability due to sequential leader–agent interactions. Together, these omissions prevent a rigorous cost–quality comparison._
>
> **Response [Cost-Quality Comparison]** We appreciate this suggestion. Our method indeed uses sequential leader–agent rounds, which we also explicitly acknowledge as a limitation in Section 5 and Appendix B.3. We have added details about the inference time required for training in the Draft (Appendix D). Below we provide GPU hours, inference tokens, and later we provide token-normalized results for each method.
>
> _GPU-Hours_: Below is a table of the total GPU hours required for training when using Qwen-2.5-7B-Instruct as the base model. We compute GPU hours as num_gpus*training_time. We have also added this to our draft (Appendix D.2, Table 13).
> | Dataset | ACC-Collab | SelectLLM | SCoRe | Deferral Leader | GRPO | MLPO
> |---------|-------------------|------------------|-------------|--------------------|----------------|----------------|
> | MMLU    | 34.2  | 18.1  | 218.3 | 17.6 | 112.3 | 172.2
> | BBH     | 21.1  | 17.2  | 127.2 | 11.1 | 82.6  | 100.5
> | MATH    | 25.7  | 15.9  | 189.9 | 18.3 | 120.5  | 168.7
>
> _Inference Tokens and Time_: Below is a table of the total time and tokens used for inference. For each dataset and method we compute the average time (number of tokens) required to complete 5 rounds of inference.
>
>
> Below we show the total number of tokens used by each method to give an answer to one question after $r=0$ and $r=4$ rounds of inference (averaged across the entire dataset). SelectLLM is computed with a budget of 20 samples.
> | Method                | MATH ($r$=$0$ / $r$=$4$) | BBH ($r$=$0$ / $r$=$4$) | MMLU ($r$=$0$ / $r$=$4$) |
> |-----------------------         |----------------|---------------|----------------|
> | Self-reflect GRPO              | 2.5k / 12k     | 2k / 9k       | 1.5k / 8k      |
> | Div MAD                        | 6k / 20k       | 4k / 14k      | 3k / 16k       |
> | SelectLLM                      | -- / 18k       | -- / 17k      | -- / 19k       |
> | ACC-Collab                     | 8k / 26k       | 5k / 18k      | 4k / 21k       |
> | Untrained leader               | 9k / 38k       | 7k / 30k      | 6k / 27k       |
> | SCoRE                          | 4k / 20k       | 3k / 12k      | 3k / 14k       |
> | MLPO Leader  (ours)            | 12k / 50k      | 10k / 40k     | 9k / 34k       |

---

> > ### Author Response · Authors · 2025-11-21
> > **Author Rebuttal (2)**
> >
> > - _I suggest that the authors can report accuracy vs. tokens/latency for varying T, K, and leader/agent sampling to characterize real‑world cost trade‑offs; include token‑normalized comparisons for baselines (extend Fig. 3/7). Moreover, they can document baseline tuning budgets; where margins are small (e.g., MATH), report sensitivity to temperature/top‑p, number of rounds, and aggregation schemes; include paired‑seed significance tests (Table 1)._
> >
> > **Response [Additional Analysis]** Thank you for the suggestions, in particular the inference time token-normalized comparisons! We have added these to our draft (Appendix B.1, Line 956, and Table 3) and also provide results below.
> >
> > _Sensitivity in slim cases_: We fully acknowledge that in some cases (for Llama and Gemma) our improvement over the baselines is slim. However, we note two things; firstly, we consistently beat all baselines (and occasionally tie), and secondly, for Qwen (a model known to be more effective with RL-based post-training) we consistently see quite large improvements over the baselines.
> >
> >
> > _Number of rounds:_ We would like to highlight that, we actually do have experiments with varying rounds in the main paper and refer the reviewer to Figure 3, 7 (full version of 3), and 9.
> >
> > _Aggregation Schemes:_ In terms of aggregation schemes, we would also like to direct the reviewer to Table 1 and Figure 3 of the main paper. Here we compare various aggregation methods, including Deferral (trained to always defer to one agent in the team), SelectLLM (from a large pool of different LLMs, decide for each input/query which subset of models to query), as well as Majority vote (Section 4.2.2).
> >
> > _Token Normalized Inference_: Below, we provide token-normalized comparisons for each method after 2 rounds of inference (rounds 0 and 1). For this experiment, each method is allowed to generate $k$ answers until the token budget is reached (if $k=0$, i.e., one answer exceeds the token budget, the answer is counted as wrong). We include our trained leader both with the agent team (MLPO Leader) and without the team (MLPO No Team). We make two important observations about these results:
> > 1. For all larger budgets (10k and above), the MLPO Leader consistently beats all baselines.
> > 2. For these larger token budget ,the MLPO leader without the team consistently beats a model trained only with GRPO.
> >
> > We acknowledge that in settings of a lower inference budget (5k tokens or less), single model solutions like GRPO may be preferable, but for larger budgets (7500 or above) our method is preferable.
> >
> > #### MMLU
> > | Token Budget | Self-Ref | Hom-MAD | MAD | Untrained Leader | GRPO | SCORE | MLPO Leader (Ours) | MLPO No Team (Ours) |
> > | :- | :-: | :-: | :-: | :-: | :-: | :-: | :-: | :-: |
> > | 2500  | 0.750$^\dagger$ | 0.414 | 0.738 | 0.364 | 0.749 | 0.757$^*$ | 0.074 | 0.704 |
> > | 5000  | 0.762 | 0.745 | 0.786$^*$ | 0.732 | 0.766 | 0.767$^\dagger$ | 0.761 | 0.763 |
> > | 7500  | 0.759 | 0.752 | 0.790$^*$ | 0.748 | 0.765 | 0.767 | 0.786$^\dagger$ | 0.769 |
> > | 10000 | 0.763 | 0.757 | 0.788$^\dagger$ | 0.755 | 0.767 | 0.767 | 0.793$^*$ | 0.771 |
> > | 15000 | 0.763 | 0.760 | 0.797$^\dagger$ | 0.758 | 0.769 | 0.767 | 0.805$^*$ | 0.773 |
> > | 20000 | 0.764 | 0.759 | 0.797$^\dagger$ | 0.765 | 0.769 | 0.767 | 0.818$^*$ | 0.773 |
> >
> >
> > ### BBH
> > | Token Budget | Self-Ref | Hom-MAD | MAD | Untrained Leader | GRPO | SCORE | MLPO Leader (Ours) | MLPO No Team (Ours) |
> > | :- | :-: | :-: | :-: | :-: | :-: | :-: | :-: | :-: |
> > | 2500  | 0.773 | 0.554 | 0.757 | 0.280 | 0.795$^\dagger$ | 0.828$^*$         | 0.026      | 0.720 |
> > | 5000  | 0.803 | 0.797 | 0.815 | 0.734 | 0.842$^*$       | 0.841$^\dagger$   | 0.749      | 0.832 |
> > | 7500  | 0.808 | 0.811 | 0.823 | 0.754 | 0.852           | 0.840             | 0.864$^*$  | 0.863$^\dagger$ |
> > | 10000 | 0.817 | 0.819 | 0.827 | 0.777 | 0.858           | 0.844             | 0.890$^*$  | 0.870$^\dagger$ |
> > | 15000 | 0.823 | 0.826 | 0.831 | 0.795 | 0.858           | 0.848             | 0.896$^*$  | 0.872$^\dagger$ |
> > | 20000 | 0.824 | 0.827 | 0.834 | 0.811 | 0.860           | 0.848             | 0.900$^*$  | 0.874$^\dagger$ |
> >
> > #### MATH
> > | Token Budget | Self-Ref | Hom-MAD | MAD | Untrained Leader | GRPO | SCORE | MLPO Leader (Ours) | MLPO No Team (Ours) |
> > | :- | :-: | :-: | :-: | :-: | :-: | :-: | :-: | :-: |
> > | 2500  | 0.705$^\dagger$ | 0.461 | 0.534 | 0.008 | 0.656           | 0.730$^*$        | 0.000     | 0.601 |
> > | 5000  | 0.726           | 0.691 | 0.685 | 0.494 | 0.735$^\dagger$ | 0.745$^*$        | 0.394     | 0.731 |
> > | 7500  | 0.737           | 0.724 | 0.696 | 0.616 | 0.761$^*$       | 0.759 $^\dagger$ | 0.648     | 0.756 |
> > | 10000 | 0.740           | 0.731 | 0.698 | 0.664 | 0.767           | 0.765            | 0.778$^*$ | 0.771$^\dagger$ |
> > | 15000 | 0.746           | 0.742 | 0.698 | 0.698 | 0.770           | 0.768            | 0.783$^*$ | 0.778$^\dagger$ |
> > | 20000 | 0.749           | 0.746 | 0.698 | 0.714 | 0.771           | 0.769            | 0.785$^*$ | 0.779$^\dagger$ |

---

### Official Review · Reviewer_v9pM · 2025-10-31

**Soundness:** 2
**Presentation:** 3
**Contribution:** 3
**Rating:** 4
**Confidence:** 4

**Summary:**

The main problem this paper aims to address is: Can we get most of the multi-agent gains by training only a single “leader” that coordinates untrained peer agents? To achieve that, this paper proposed Multi‑agent guided Leader Policy Optimization (MLPO), SFT to instill backtracking/self‑correction followed by a GRPO‑style RL phase to train a leader. Empiracally, this algorithm outperforms strong single‑ and multi‑agent baselines on BBH, MATH, and MMLU (Table 1, p. 6), and the leader is even stronger alone (zero‑shot, without the team) than standard GRPO‑trained models (§4.2.3; Fig. 4, p. 7).

**Strengths:**

-- Clear, modular architecture. §3.1 (pp. 3–4) details a two‑level hierarchy and the T‑round interaction loop; Figure 1 (p. 3) visually clarifies the leader–agents workflow and the think/answer structure the leader emits.

-- Well‑specified objective. §3.2 (pp. 4–5) formalizes MLPO as a GRPO variant that conditions the leader on agent responses, with Dr.GRPO‑style stability tweaks; the training‑data pipeline (4K agent proposals per task; filtered “easy” tasks) is explicit (§3.2, p. 5).

-- Robustness to weak agents. Figure 5 (p. 8) and Figure 10 (p. 20) demonstrate the trained leader can override incorrect team suggestions—performance holds up even when few/none of the agents are correct.

-- The claim"Leaders trained with MLPO exhibit improved performance not only when interacting with the agent team at inference time, but also enjoy improved performance when deployed in single-agent settings without the team" is an insight for me.

**Weaknesses:**

The most concern for me is the potential unfair comparison:

-- SFT transparency & potential advantage. Appendix A.1 describes how the SFT data are constructed (synthetic backtracking/self‑correction), but omits crucial statistics: dataset size, token counts, domain/source mix, and sampling rules (p. 17). This makes it hard to judge how much of the gain stems from SFT itself vs. MLPO, and it obscures fairness vs. baselines that may not receive equivalent SFT.

-- Uneven pretraining across baselines. The main text and Appendix D list training‑based baselines and their hyperparameters, but do not state that baselines like ACC‑Collab, SCoRe, GRPO, or SelectLLM were also given an equally sized, distribution‑matched SFT/backtracking set; only the Deferral Leader is said to be trained on “the same data” as MLPO (Appendix D, p. 25). This risks a pro‑MLPO bias.

-- Budget inconsistency at inference. §4.2.2 claims parity via “at most 40 total generations” for majority vote (p. 7), but Appendix D configures SelectLLM with a 20‑vote budget and says this matches “the 20 total inference‑time generations used by our pipeline for 5 rounds of inference” (p. 25). Which one is actual setting?

-- Protocol advantages not fully controlled. The method uses structured prompts and shares rich reasoning traces between team and leader (Appendix E.1–E.2, pp. 25–27). Some baselines (e.g., SelectLLM) primarily select models rather than aggregate intermediate reasoning, and may not receive comparable intermediate evidence, making the comparison sensitive to input protocol rather than algorithmic merit.

**Questions:**

Besides the weakness, i have some problem to discuss with the author with respect:
-- I am wondering the training compute, can you provide GPU‑hours, effective training tokens, #steps/epochs, and KL/regularization schedules so readers can assess budget parity (Appendix B.3, p. 23; Appendix D, pp. 24–25)? Much thanks!

-- Effect of filtering “easy” tasks. Since §3.2 filters tasks where ≥75% of 4K agent responses are correct (p. 5) and Table 6 (p. 23) shows benefits, were analogous difficulty controls applied to baselines? If not, can you provide results without filtering to isolate MLPO’s contribution?

-- Zero‑shot comparison tokens. In §4.2.3 (Fig. 4, p. 7), does the Zero‑shot GRPO baseline see the same total training tokens as the SFT+MLPO leader? If not, can you provide a token‑matched comparison?

---

> ### Author Response · Authors · 2025-11-21
> **Author Rebuttal (1)**
>
> - _SFT transparency & potential advantage. Appendix A.1 describes how the SFT data are constructed (synthetic backtracking/self‑correction), but omits crucial statistics: dataset size, token counts, domain/source mix, and sampling rules (p. 17). This makes it hard to judge how much of the gain stems from SFT itself vs. MLPO, and it obscures fairness vs. baselines that may not receive equivalent SFT._
>
>
> **Response [MLPO gains from SFT]** We agree that isolating the effects of SFT on MLPO is important! This is precisely why we provide results for pure SFT, MLPO without SFT, and MLPO with SFT in our main results (Table 1). In this table, the row "SFT Leader" refers to a leader trained only on the SFT data, "SFT + MLPO Leader" refers to MLPO with the initial SFT step, and "MLPO Leader" refers to MLPO without this initial SFT. We find that while this SFT step increases performance, the majority of the performance is obtained from MLPO itself, for example, on BBH the untrained leader attains accuracy 0.764, the SFT leader the MLPO leader attains 0.865, and SFT + MLPO Leader attains 0.882.
>
>
>
> **Response [Sampling SFT data]**
> We described SFT sampling in Appendix E.3 of our draft. However, in light of this question, we have added additional clarifications to this Appendix E.3, and added details in Appendix D.3. For completeness we reiterate these details here. To teach the leader model natural backtracking and self-correction behavior, we construct a supervised fine-tuning (SFT) dataset comprising synthetic completions that mimic realistic reasoning failures followed by recovery.
>
> For each dataset, we begin by running our untrained multi-agent pipeline to collect 16 distinct completions from the leader for each agent team generation. We discard examples where all completions are incorrect or correct. For the remaining examples, we randomly select one correct and one incorrect leader completion. These are then used to construct a prompt that asks an untrained leader model to imitate a reasoning process that starts with plausible but flawed logic (using the incorrect trajectory), then backtracks and self-corrects to reach the correct answer (using the correct trajectory). We then filter out any remaining examples where the backtracking did not yield a correct final answer. The generated backtracked completions form the core of our SFT dataset. We have also included an example of the backtracked completion used for SFT in Appendix E.3. We use an identical process for GRPO, without the agent team involvement and with a higher sample budget to ensure a similar number of examples are used for both methods (32).
>
>
>
> **Response [Training Statistics]**
> Below, we have added a table outlining the SFT training data statistics for both MLPO and GRPO in our paper. Note that these are the only two methods that receive an initial SFT step (SCoRE has its own two-phase training, ACC-Collab combines SFT and DPO during training). For both MLPO and GRPO, the training data has roughly the same number of training examples. We note that while the prompt length of the SFT data is much larger for MLPO (which is masked in the loss), the response length is fairly similar (which is not masked in the loss). This table is also added to our draft (Table 15).
>
>
> **Average Token Length**
> | Dataset | Training Examples (GRPO/MLPO) | GRPO Prompt | MLPO Prompt | GRPO Response | MLPO Response |
> |---------|------------------|------------------|-------------|--------------------|---------------|
> | MMLU    | 7,431 / 7,470  | 185.7   | 976.3       | 1650.4             | 1846.3        |
> | BBH     | 2,211 / 2,192  | 189.7   | 1072.5      | 1430.6             | 1665.5        |
> | MATH    | 4,603 / 4,744  | 204.3   | 1301.3      | 1488.8             | 1552.5        |

---

> ### Author Response · Authors · 2025-11-21
> **Author Rebuttal (2)**
>
> - _Uneven pretraining across baselines. The main text and Appendix D list training‑based baselines and their hyperparameters, but do not state that baselines like ACC‑Collab, SCoRe, GRPO, or SelectLLM were also given an equally sized, distribution‑matched SFT/backtracking set; only the Deferral Leader is said to be trained on “the same data” as MLPO (Appendix D, p. 25). This risks a pro‑MLPO bias._
>
> **Response [Details on pretraining across Baselines]** We apologize for the lack of clarity on this point. We have added these details to the updated version of our draft (Appendix D.1 and D.3). For clarity, we also provide them here. For GRPO, we use an identical setup, i.e., an initial SFT step with backtracking data (using the same number of examples as MLPO) followed by an RL step. SCoRE also utilizes an initial training step (critique/refine format rather than backtracking) which uses all questions in the training set (more than our SFT set). ACC-Collab does not make use of such a pre-training set, since this method uses a mix of SFT and DPO and requires a specific data format during training. The deferral leader and SelectLLM are also trained on these examples (in additional the same samples that the MLPO leader sees during its GRPO training), however these baselines are not trained on the completions since they are classifier (last FC 2 layers of the LLM retrained), which either select the agent response most likely to be correct (deferral leader) or selects which subset of LLMs should be queried for majority vote aggregation (SelectLLM).
>
> ---
> - _Budget inconsistency at inference. §4.2.2 claims parity via “at most 40 total generations” for majority vote (p. 7), but Appendix D configures SelectLLM with a 20‑vote budget and says this matches “the 20 total inference‑time generations used by our pipeline for 5 rounds of inference” (p. 25). Which one is actual setting?_
>
> **Response [SelectLLM Budget]** We apologize for the confusion. SelectLLM is unique compared to the other baselines since it is the only baseline with majority-vote baked in; it intakes a sample budget $k$, question $q$ and collection of models $M_1, \ldots, M_n$, and determines how to distribute the sample budget over the $n$ models. For a fair comparison, we set $k$ to the total number of samples used by our method in our two settings. For all results in Section 4.2.2 (where we investigate sample-normalized majority vote) SelectLLM uses a 40-vote budget in order to match the total sample budget used by all methods in this section. Outside of Section 4.2.2, SelectLLM is given a budget of 20 total samples in order to match the number of samples used by our method (3 agents + 1 leader, over 5 rounds).  We have clarified this in the updated version of our draft (Appendix D.1 Line 1428).
>
> ---

---

> > ### Author Response · Authors · 2025-11-21
> > **Author Rebuttal (3)**
> >
> > - _Protocol advantages not fully controlled. The method uses structured prompts and shares rich reasoning traces between team and leader (Appendix E.1–E.2, pp. 25–27). Some baselines (e.g., SelectLLM) primarily select models rather than aggregate intermediate reasoning, and may not receive comparable intermediate evidence, making the comparison sensitive to input protocol rather than algorithmic merit._
> >
> > **Response [Protocol of Hierarchical Inference]**
> > This is a great point! Yes, some baselines are limited by their own protocol, as you point out. SelectLLM fundamentally lacks "intermediate reasoning", although other baselines like ACC-Collab and SCoRe are necessarily based on this intermediate reasoning. We view this as an advantage of our method over methods like SelectLLM, which operate more like a dynamic ensemble; the totality of our training pipeline and inference pipeline allows the leader to better leverage all available information, this is one aspect of what gives MLPO its superior performance
> >
> >
> > When evaluating our trained models, we were also curious about the effects of information sharing, which is why we provided an analysis on the effects of information shared between the leader and the team in Figure 11.  We provide the final round performance of each of these different variations of MLPO below. When comparing these results with the baselines in Table 1, we see that while sharing full information between the agent team and leader is typically best, even when agents only share their final answer (no reasoning) MLPO still beats all baselines aside from two cases (diverse MAD and  SelectLLM on MMLU).
> > | Dataset | Agents Give Answers Only | Agents Given Reasoning Only | Leader Gives Answers Only | Full Information |
> > |:---------:|:---------------------------:|:------------------------------:|:----------------------------:|:-------------------:|
> > | MMLU    | 0.767                     | 0.790                        | 0.778                      | 0.782             |
> > | BBH     | 0.872                     | 0.847                        | 0.868                      | 0.882             |
> > | MATH    | 0.759                     | 0.762                        | 0.751                      | 0.762             |
> >
> > Furthermore, the zero-shot performance of our leader (i.e., the leader with no team and no multi-round inference) also exhibits strong performance. In Table 1 we see that no single agent baseline beats our leader when deployed as single agent, and across all baselines the zero-shot deployment of our leader beats all baselines aside from four cases (diverse/homogeneous MAD, ACC-Collab, and SelectLLM on MMLU), despite the fact that three of those baselines (both MAD versions and ACC-Collab) use multi-step inference and multiple models.
> >
> > While we view the protocol (sharing of rich information) of MLPO as an advantage, these findings suggest that it cannot fully explain the success of MLPO.
> >
> > - _Besides the weakness, i have some problem to discuss with the author with respect: -- I am wondering the training compute, can you provide GPU‑hours, effective training tokens, #steps/epochs, and KL/regularization schedules so readers can assess budget parity (Appendix B.3, p. 23; Appendix D, pp. 24–25)? Much thanks!_
> >
> > **Response [Training Details]** We appriciate the throughness. We have added these details in the updated version of our pdf (Appendix D). We also provide them here:
> >
> > _KL Regularization_: We keep the KL penality (denoated as $\beta$) constant.
> >
> > _Effective Training Tokens_: To the best of our knowledge, this metric is not commonly reported. We are not aware of any other multi-agent training paper that makes such a comparison, and it is our understanding that even other GRPO-based training works, such as DAPO and Dr.GRPO do not report effective training tokens.
> >
> > _epochs_ We use 1 epoch for GRPO and MLPO, 4 for ACC-Collab, 2 for SCoRE, 50 for SelectLLM and Deferral Leader (these are classifiers on LLM hidden states). Number of steps will vary depending on dataset (since different numbers of examples are used).
> >
> > _GPU-Hours_: Below is a table of the total GPU hours required for training when using Qwen-2.5-7B-Instruct as the base model. We compute GPU hours as num_gpus*training_time. This table is added to our draft (Table 13).
> > | Dataset | ACC-Collab | SelectLLM | SCoRe | Deferral Leader | GRPO | MLPO
> > |---------|-------------------|------------------|-------------|--------------------|----------------|----------------|
> > | MMLU    | 34.2  | 18.1  | 218.3 | 17.6 | 112.3 | 172.2
> > | BBH     | 21.1  | 17.2  | 127.2 | 11.1 | 82.6  | 100.5
> > | MATH    | 25.7  | 15.9  | 189.9 | 18.3 | 120.5  | 168.7

---

> > > ### Author Response · Authors · 2025-11-21
> > > **Author Rebuttal (4)**
> > >
> > > - _Effect of filtering “easy” tasks. Since §3.2 filters tasks where ≥75% of 4K agent responses are correct (p. 5) and Table 6 (p. 23) shows benefits, were analogous difficulty controls applied to baselines? If not, can you provide results without filtering to isolate MLPO’s contribution?_
> > >
> > > **Response [Difficulty Filtering and Baselines]** We apply the same filtering to the RL baselines, namely GRPO and SCoRE, but not to the other baselines as it results in worse performance for the other baselines. Moreover, some baselines, like ACC-Collab, have their own filtering scheme.
> > >
> > > The filtering has a two-fold benefit for MLPO. Firstly, since MLPO uses GRPO-style training, "easy" questions will frequently yield an advantage term of 0 (since the reward for each rollout is identical). Secondly, "easy" questions have the potential to teach the leader a degenerate signal of simply agreeing with the majority of agents. We are not sure why the filtering results in worse performance for baselines like ACC-Collab and SelectLLM although this is beyond the scope of our work.
> > >
> > >
> > >
> > > - _Zero‑shot comparison tokens. In §4.2.3 (Fig. 4, p. 7), does the Zero‑shot GRPO baseline see the same total training tokens as the SFT+MLPO leader? If not, can you provide a token‑matched comparison?_
> > >
> > > **Response [Token Normalized Training vs Sample Normalized Training]** To the best of our knowledge, token-normalized training comparisons are not common in multi-agent training papers (we are not aware of a single multi-agent training paper that makes such a comparison), or even in GRPO-based training papers such as [DAPO](https://arxiv.org/pdf/2503.14476) or [Dr.GRPO](https://arxiv.org/pdf/2503.20783), which provides their comparisons in terms of the number of training steps (MLPO and GRPO share the same number of training steps in our setting). In our experiment,s both MLPO and GRPO expose the model to the same number of examples (resulting in the same number of updates). However, MLPO almost certainly exposes the model to more tokens; this is a key advantage of MLPO, not an asymmetry of our experimental comparisons. Because MLPO uses the joint of questions q and agents' solutions S, the data domain can be far richer than using the questions alone. MLPO is implicitly a data augmentation method that allows multiple training samples $(q, S_1)$, $(q, S_2)$, etc, to stem from the same question q by leveraging different agent solutions $S_i$.

---

> > > > ### Comment · Reviewer_v9pM · 2025-11-28
> > > > **Answer acknowledged**
> > > >
> > > > Thank the authors for the detailed rebuttal and for providing the requested statistics on compute and training data. While I appreciate the transparency, the new information reinforces my initial concerns regarding the source of the performance gains and the efficiency of the proposed method.
> > > >
> > > > I have decided to maintain my score based on the following two key reasons:
> > > >
> > > > 1. "Implicit Data Augmentation" vs. Algorithmic Novelty
> > > > In response to the query about token count fairness, the authors state that "MLPO almost certainly exposes the model to more tokens" and characterize this as a benefit of "implicit data augmentation."To me, this confirms that the performance advantage may not stem from the superior design of the optimization mechanism (the RL/GRPO component), but simply from the fact that the model consumes significantly more synthetic reasoning tokens from agents during training than the baselines. This aligns more with the paradigm of computation-heavy Knowledge Distillation or extensive Data Augmentation rather than a novel Multi-Agent RL contribution. If the gain disappears when baselines are allowed to "read" an equivalent volume of synthetic reasoning traces (e.g., via simple SFT or DPO on the same agent outputs), then the value of the complex MLPO reinforcement learning loop is diminished.
> > > >
> > > >
> > > > 2. Marginal Gains vs. High Complexity
> > > > The rebuttal breakdown reveals that the "SFT Leader" alone already achieves strong performance (e.g., 0.865 on BBH), while the full "SFT + MLPO Leader" improves this to 0.882.This yields a marginal gain of $\sim1.7\%$. However, achieving this requires a highly complex and computationally expensive pipeline: Generation (4K proposals) $\rightarrow$ Filtering $\rightarrow$ SFT Construction $\rightarrow$ SFT Training $\rightarrow$ GRPO RL Training.
> > > >
> > > > Given the high GPU-hours reported in the rebuttal (e.g., MLPO requires 172.2h on MMLU vs. significantly less for inference-time baselines or simpler training), the cost-benefit ratio is not favorable. The method appears to be an inefficient way to squeeze out small performance gains compared to the robust SFT baseline.
> > > >
> > > > Suggestions for Future Improvement:
> > > >
> > > > To clearly demonstrate the value of MLPO in future revisions, I suggest the following:
> > > >
> > > > Token-Matched Baseline: The most critical missing piece is a baseline that is trained on the same volume of agent-generated tokens but using a simpler objective (e.g., standard SFT or DPO on the correct agent trajectories without the RL loop). This would isolate whether the gain comes from the training signal (MLPO) or just the data scale.
> > > >
> > > > Efficiency Analysis: Investigate if the pipeline can be simplified. For instance, can the initial SFT phase be removed, or can the number of agent proposals needed for the RL phase be drastically reduced? Showing that MLPO works efficiently with a smaller "token budget" would strengthen the algorithmic claim.
> > > >
> > > > Disentangling Reasoning vs. Aggregation: The paper claims the leader learns to "coordinate." However, since it sees full reasoning traces, it might just be learning to select the longest/most complex answer. An analysis on whether the leader is actually performing reasoning aggregation versus simple heuristic selection would add depth to the analysis.

---

> > > > > ### Author Response · Authors · 2025-12-03
> > > > > **Response to Reviewer**
> > > > >
> > > > > - _"Implicit Data Augmentation" vs. Algorithmic Novelty In response to the query about token count fairness, the authors state that "MLPO almost certainly exposes the model to more tokens" and characterize this as a benefit of "implicit data augmentation."To me, this confirms that the performance advantage may not stem from the superior design of the optimization mechanism (the RL/GRPO component), but simply from the fact that the model consumes significantly more synthetic reasoning tokens from agents during training than the baselines. This aligns more with the paradigm of computation-heavy Knowledge Distillation or extensive Data Augmentation rather than a novel Multi-Agent RL contribution. If the gain disappears when baselines are allowed to "read" an equivalent volume of synthetic reasoning traces (e.g., via simple SFT or DPO on the same agent outputs), then the value of the complex MLPO reinforcement learning loop is diminished._
> > > > >
> > > > >
> > > > > **Response [Algorithmic Novelty]** A key advantage of MLPO is that it allows multiple training examples to stem from the same question, thus allowing us to extend our training data so long as we can generate possible agent solutions. This ability to get more out of a given fixed training set is a key advantage of MLPO. The way in which MLPO extends these examples is itself a key novelty; for each question $q$, MLPO exposes the leader to $k$ different possible solution sets $S_1, …, S_k$, yielding examples $(q, S_1), …, (q, S_k)$. This exposure to multiple possible solutions allows the leader to better explore the space of solutions at training time. This goes beyond mere token exposure and represents a novel data augmentation that not only improves the model’s performance as a leader, but also as a single model functioning in a zero-shot setting with no team (see Table 1).
> > > > >
> > > > >
> > > > > - _Marginal Gains vs. High Complexity The rebuttal breakdown reveals that the "SFT Leader" alone already achieves strong performance (e.g., 0.865 on BBH), while the full "SFT + MLPO Leader" improves this to 0.882.This yields a marginal gain of . However, achieving this requires a highly complex and computationally expensive pipeline: Generation (4K proposals)  Filtering  SFT Construction  SFT Training  GRPO RL Training._
> > > > >
> > > > >
> > > > > **Response [Gains from SFT are minimal]** There appears to be some confusion so we have provided a table quoting the performance, from our paper, of the untrained leader, MLPO with SFT-only leader, MLPO with RL-only leader, and MLPO with an SFT+RL leader (our main method). In the table below we see that the performance of the SFT-only leader is almost equivalent to that of the untrained leader, while the performance, **implying that the gains from SFT are minimal, and that the bulk of the improvement comes from the RL-stage of MLPO**.
> > > > >
> > > > >
> > > > > For example, in the table below, the SFT-only leader has identical performance with the untrained leader on BBH, while the MLPO-only leader has 10% higher accuracy.
> > > > >
> > > > >
> > > > > | | MMLU | BBH | MATH |
> > > > > |--|--|--|--|
> > > > > Untrained Leader                   |  0.731 | 0.764 | 0.697 |
> > > > > MLPO with SFT-Only Leader | 0.741 | 0.762 | 0.713 |
> > > > > MLPO with RL-Only Leader  | 0.759 | 0.865 | 0.750 |
> > > > > MLPO with SFT+RL Leader   | 0.782 | 0.882 |  0.762 |
> > > > >
> > > > >
> > > > >
> > > > >
> > > > > - _Given the high GPU-hours reported in the rebuttal (e.g., MLPO requires 172.2h on MMLU vs. significantly less for inference-time baselines or simpler training), the cost-benefit ratio is not favorable. The method appears to be an inefficient way to squeeze out small performance gains compared to the robust SFT baseline._
> > > > >
> > > > > **Response [Training Time]** While MLPO requires more GPU hours for training, the increased performance is clear: MLPO beats, or occasionally ties with, all training baselines. We agree that for some users the increased training time may not be acceptable, but for those with additional compute, MLPO offers a straightforward means of boosting performance.
> > > > >
> > > > >
> > > > > - _Token-Matched Baseline: The most critical missing piece is a baseline that is trained on the same volume of agent-generated tokens but using a simpler objective (e.g., standard SFT or DPO on the correct agent trajectories without the RL loop). This would isolate whether the gain comes from the training signal (MLPO) or just the data scale._
> > > > >
> > > > >
> > > > > **Response [Token-Matched Baseliness]**  As mentioned previously, training with equal numbers of tokens is not a common evaluation metric. We are not aware of any similar works that make token matched comparisons (e.g., DAPO and Dr. GRPO both do not provide this evaluation metric). It would appear that the reviewer is also not aware of any such works, as they declined to provide any works that make this comparison.

---

> > > > > > ### Author Response · Authors · 2025-12-03
> > > > > > **Response to Reviewer (2)**
> > > > > >
> > > > > > - _Efficiency Analysis: Investigate if the pipeline can be simplified. For instance, can the initial SFT phase be removed, or can the number of agent proposals needed for the RL phase be drastically reduced? Showing that MLPO works efficiently with a smaller "token budget" would strengthen the algorithmic claim._
> > > > > >
> > > > > > **Response [Simplifying the pipeline and removing SFT]** While we thank the reviewer for this suggestion, we will point out again that **this suggested experiment is already part of our work and is one of the first results presented in the paper**. As outlined in Table 1, we find that the majority of the increased performance in MLPO comes from the RL-phase, not the SFT-phase; see SFT Leader (SFT-only)  vs MLPO Leader (RL-only) vs SFT+MLPO Leader (SFT + RL). If one does not want to use the initial SFT step, they can still get nearly all of the performance gains from using only the RL step.
> > > > > >
> > > > > >
> > > > > > - _Disentangling Reasoning vs. Aggregation: The paper claims the leader learns to "coordinate." However, since it sees full reasoning traces, it might just be learning to select the longest/most complex answer. An analysis on whether the leader is actually performing reasoning aggregation versus simple heuristic selection would add depth to the analysis._
> > > > > >
> > > > > >
> > > > > > **Response [Disentangling Reasoning vs. Aggregation]** We also thank the reviewer for this suggestion as well, but we again point out that **these suggested experiments are also already in our paper**. As mentioned previously, we were also curious whether the leader is merely performing aggregation, so we conducted three experiments to test this.
> > > > > > 1. We test the leader’s ability when no team is present (i.e., it is impossible to do aggregation). In Figure 4 we show that the leader has improved performance as a single model without the team compared to a single model trained with GRPO.
> > > > > > 2. We ablate the effects of providing the leader with a) agent reasoning only, b) agent answers only, and c) full information. We find that while providing full information is typically best, agents providing only answers or only reasoning still allow the leader to have high performance. This indicates that the leader is not simply following a trivial signal such as the longest reasoning or the most complex answer.
> > > > > > 3. Lastly, we compare to a deferral leader designed to select the best agent response from the team. In Table 1 we see that the MLPO leader significantly outperforms this baseline.

---

### Official Review · Reviewer_E79c · 2025-10-31

**Soundness:** 3
**Presentation:** 3
**Contribution:** 3
**Rating:** 4
**Confidence:** 3

**Summary:**

This paper proposes a hierarchical multi-agent framework that trains a single leader LLM to coordinate several untrained companion agents, thereby reducing training costs while improving collaborative reasoning performance. Specifically, the paper introduces MLPO (Multi-agent guided Leader Policy Optimization). It first constructs a set of examples featuring self-correction/backtracking and uses SFT to enhance the leader’s abilities in aggregation, error correction, and backtracking. Then, within the GRPO framework, it treats the agents’ solutions as additional training context to directly optimize the leader model’s capability to evaluate and synthesize multiple answers. Experimental results show that, in multi-agent LLM systems, training only a flexible leader enables efficient and effective collaborative reasoning. Moreover, the trained leader model also achieves performance gains when deployed as a single model.

**Strengths:**

1. Training only a single leader LLM to cooperate among several untrained companion agents significantly reduces training and maintenance costs, yet preserves the benefits of collaboration.
2. The paper introduces the MLPO training framework: construct SFT data to enhance the model’s self-correction ability, and directly optimize the leader under the GRPO framework—resulting in a relatively simple training process.
3. The trained leader model can operate in both “team collaboration” and “single-model” modes, indicating that the learned aggregation/correction strategies transfer to general reasoning ability.

**Weaknesses:**

1. The experiments are conducted with a collaboration setting of K = 3 agents, without verifying the effectiveness of the proposed method as the number of untrained agents increases. Since MLPO treats untrained agents as part of the environment, increasing their number would make the environment more complex.
2. Although training only a single leader model reduces training costs, the overall collaboration quality may be constrained by the leader’s capability. If the leader LLM’s task ability is weak, it could perform worse than untrained agents debating directly or a single untrained agent answering alone.
3. During training, “easy tasks” (those with a team accuracy ≥ 75%) are filtered out to focus on the leader’s coordination skills. This 75% threshold is heuristic and lacks theoretical grounding or systematic validation.
4. It remains unclear whether the trained leader exhibits strong cross-team and cross-task generalization. For example, can a leader trained with one team configuration effectively guide a different team of agents? Can a leader trained on one task domain provide strong leadership on related tasks?

**Questions:**

See weaknesses.

**Details Of Ethics Concerns:**

The paper raises no ethical concerns.

---

> ### Author Response · Authors · 2025-11-21
> **Author Rebuttal (1)**
>
> - _It remains unclear whether the trained leader exhibits strong cross-team and cross-task generalization. For example, can a leader trained with one team configuration effectively guide a different team of agents? Can a leader trained on one task domain provide strong leadership on related tasks?_
>
> **Response [Generalization]** We thank the reviewer for this thought-provoking question! As a result, we ran a few additional experiments testing our leader's ability to generalize to other teams. Below we provide a table of results for our trained leader (trained on the Qwen + Gemma + Llama team) when paired with homogeneous teams and with no team (equivalent to the zero-shot leader results presented in the paper). We mark the highest result with * and the second highest with $^\dagger$. We have also added these to the updated version of our draft (Appendix B.2 Line 1178, and Table 6).
> We make two observations regarding these results:
> 1. The trained leader exhibits a strong ability to generalize from a heterogeneous team to each of the homogeneous teams.
> 2. In some cases, the leader trained on a heterogeneous team, paired with a homogeneous team at inference time, can outperform a leader trained on that homogeneous team (Figure 12), e.g., the Qwen team on BBH. We believe this gives more evidence for our hypothesis that exposing the leader to diverse solutions at inference is a key element to the leader's success (solutions will be naturally more diverse with a heterogeneous team).
>
>
>
> | Agent team  | MMLU |  BBH | MATH |
> |----------|----------|----------|----------|
> | Qwen + Gemma + Llama (seen)|  0.782* | 0.882* | 0.762*                   |
> |||||
> | 3x Qwen  (unseen)  | 0.760           | 0.858           | 0.761$^\dagger$ |
> | 3x Gemma  (unseen) | 0.762$^\dagger$ | 0.868$^\dagger$ | 0.701           |
> | 3x Llama  (unseen) | 0.745           | 0.862           | 0.680           |
> |No Team (unseen)    | 0.757           | 0.855           | 0.729           |
> ||||||
> |Untrained Leader (baseline) | 0.731 | 0.764 |  0.697 |
> |GRPO (baseline)             | 0.742 | 0.791 |  0.712 |
>
>
> In addition to the new results above, we also want to highlight that we do show the effectiveness of MLPO under different leaders (Appendix C) and agent team compositions (Appendix B.2 and Figure 12), although for each combination, the leader is trained with that team. Additionally, we show that the leader exhibits strong zero-shot performance (Figure 4), i.e., generalizing to the case of no agent team (note that the leader sees no zero-shot prompts at training time).
>
> Lastly, we agree that generalization to unseen domains is important to study, however, we leave this to future work; given that our current trained leader models are trained for each domain (as are the baselines), we would need to retrain these leaders (and baselines) on mixture datasets (e.g., MATH + BBH and then evaluate on MMLU) in order to properly test out of domain generalization.

---

> > ### Author Response · Authors · 2025-11-21
> > **Author Rebuttal (2)**
> >
> > - _The experiments are conducted with a collaboration setting of K = 3 agents, without verifying the effectiveness of the proposed method as the number of untrained agents increases. Since MLPO treats untrained agents as part of the environment, increasing their number would make the environment more complex._
> >
> > **Response [Use of 3 Agents]** We agree that understanding performance as the number of agents increases is important. Our choice of K = 3 was based on initial experiments indicating that this setting gives a good balance between diversity and compute cost (more agents means more tokens and longer run time). Although we study 3 models, we want to be clear that the method itself is agnostic to the number of agents. We have included a set of results to show that MLPO is still effective with a varying number of agents. To isolate the effect of the number of agents, rather than _which_ agents are in the team, we conduct experiments with homogeneous agent teams. These results are an extension of the above.
> >
> > We have added these results to the updated version of our darft (Appendix B.2, Line 1222, and Table 7)
> >
> > Below we show results when using our trained leader (trained on a heterogeneous team) vs the untrained leader on teams consisting of $K$ agents for $0 \leq K \leq 6$. We make two observations on these results:
> > 1. The trained leader is quite robust to the choice of team, both in terms of the number of agents and the composition of those agents.
> > 2. In some cases of particularly weak teams (e.g., Llama teams on MATH), it is better to deploy the leader solo, i.e., sometimes $K=0$ is best and performance continues to drop as $K$ increases. This lends more support to our hypothesis that diverse teams are generally the most effective (as shown in Figure 12).
> >
> >
> >
> >
> > #### MMLU
> > | Agent Team | 0 (No Team) | 1 | 2 | 3 | 4 | 5 | 6 |
> > | :- | :-: | :-: | :-: | :-: | :-: | :-: | :-: |
> > | Qwen-2.5-7B-Instruct (trained/untrained)       | 0.757/0.746 | 0.757/0.698 | 0.758/0.723 | 0.760/0.720 | 0.761/0.722 | 0.752/0.699 | 0.748/0.681 |
> > | Gemma-2-9b-it (trained/untrained)              | 0.757/0.746 | 0.760/0.741 | 0.762/0.751 | 0.762/0.757 | 0.759/0.756 | 0.757/0.750 | 0.750/0.743 |
> > | Meta-Llama-3.1-8B-Instruct (trained/untrained) | 0.757/0.746 | 0.752/0.729 | 0.750/0.730 | 0.745/0.738 | 0.743/0.727 | 0.741/0.732 | 0.737/0.719 |
> >
> >
> > #### BBH
> > | Agent Team | 0 (No Team) | 1 | 2 | 3 | 4 | 5 | 6 |
> > | :- | :-: | :-: | :-: | :-: | :-: | :-: | :-: |
> > | Qwen-2.5-7B-Instruct   (trained/untrained)     | 0.855/0.766 | 0.856/0.753 | 0.856/0.750 | 0.858/0.759 | 0.856/0.760 | 0.853/0.763 | 0.849/0.747 |
> > | Gemma-2-9b-it         (trained/untrained)      | 0.855/0.766 | 0.858/0.751 | 0.866/0.751 | 0.868/0.750 | 0.865/0.750 | 0.859/0.755 | 0.843/0.741 |
> > | Meta-Llama-3.1-8B-Instruct (trained/untrained) | 0.855/0.766 | 0.860/0.755 | 0.860/0.751 | 0.862/0.755 | 0.857/0.756  | 0.840/0.765 | 0.839/0.763 |
> >
> >
> > #### MATH
> > | Agent Team | 0 (No Team) | 1 | 2 | 3 | 4 | 5 | 6 |
> > | :- | :-: | :-: | :-: | :-: | :-: | :-: | :-: |
> > | Qwen-2.5-7B-Instruct       (trained/untrained) | 0.729/0.681 | 0.758/0.755 | 0.756/0.751 | 0.761/0.759 | 0.755/0.754 | 0.758/0.762 | 0.756/0.754  |
> > | Gemma-2-9b-it              (trained/untrained) | 0.729/0.681 | 0.710/0.619 | 0.703/0.613 | 0.701/0.602 | 0.654/0.589 | 0.644/0.579 |  0.601/0.580 |
> > | Meta-Llama-3.1-8B-Instruct (trained/untrained) | 0.729/0.681 | 0.719/0.590 | 0.682/0.598 | 0.680/0.599 | 0.677/0.596 | 0.669/0.561 | 0.654/0.50 |

---

> ### Author Response · Authors · 2025-11-21
> **Author Rebuttal (3)**
>
> **Response [Training with differing team sizes]**
> We have run additional results for MLPO  applied to different team sizes, namely 2 and 4 agents (results for 3 agents are from our original draft). We ran these experiments on homogenous teams consisting of Gemma, Qwen, or Llama, since altering the number of agents with diverse teams introduces the confounder of _which_ model to remove or add.
> Results are provided in the table below and in the draft (Appendix B.2, Line 1168, and Table 5); we make three observations about these results:
> 1. Performance is fairly stable for these two additional team sizes.
> 2. Although there is a general trend that using more models (4 vs 3 or 2) is better, however we would expect to see performance drop after a certain point. Based the above experiments, we speculate that this decrease would likely occur around 5 or 6 agents.
> 3. A diverse team still appears to be the best choice as it either beats, or ties with, each of the homogeneous teams, even when that team has an additional agent.
> #### MMLU
> | Team        | 2   | 3           | 4   |
> |-------------|-----|-------------|-----|
> | Gemma Team  |    0.759 | 0.765    |  0.776   |
> | Qwen Team   |   0.744  | 0.741    |  0.745   |
> | Llama Team  |    0.740 | 0.752    |  0.758   |
> | Diverse Team|   -  | 0.782    |   -  |
>
>
> #### BBH
> | Team        | 2   | 3           | 4   |
> |-------------|-----|-------------|-----|
> | Gemma Team  |   0.833 | 0.840    |  0.842   |
> | Qwen Team   |   0.839  | 0.812    |  0.842    |
> | Llama Team  |   0.861  | 0.878    |   0.869  |
> | Diverse Team|  -   | 0.882    |   -  |
>
>
> #### MATH
> | Team        | 2   | 3           | 4   |
> |-------------|-----|-------------|-----|
> | Gemma Team  |   0.730  | 0.744    |  0.746   |
> | Qwen Team   |   0.741  | 0.762    |   0.765  |
> | Llama Team  |   0.705  | 0.697    |  0.690   |
> | Diverse Team|  -   | 0.762    |   -  |
>
>
>
> ---
> - _Although training only a single leader model reduces training costs, the overall collaboration quality may be constrained by the leader’s capability. If the leader LLM’s task ability is weak, it could perform worse than untrained agents debating directly or a single untrained agent answering alone._
>
> **Response [Sensitivity to choice of leader]** We agree that MLPO's performance depends on the strength of the leader model; we actually highlight this point in Section 4.4 and Appendix C. This sensitivity is not unique to MLPO, as any post-training-based multi-agent method (ACC-Collab, SCoRe, SelectLLM, etc.) will exhibit the same dependency. That said, we note that the dependence may be more significant in our case since the leader is in a "centralized" role and thus has a greater influence on the process. Importantly, across all three benchmarks, MLPO still consistently outperforms baselines (Tables 1, 7, 8); in two cases where our method struggles to beat untrained agents (namely Gemma and Llama leaders on MMLU), no training-based baseline can outperform untrained diverse MAD (Appendix C, Tables 7, 8). We also note that the flip side of this sensitivity is true; stronger leaders would also be more robust to a weak agent team (similar to observations made in Figure 5).
>
> We fully agree that this is an important point to be made more explicitly and thus we have added the following to our conclusion on line 497:
> > Further, MLPO relies on a central leader model, and is more sensitive to the leader's efficacy (compared with other multi-agent approaches that do not utilize a central agent); as such, we recommend using a model with comparable strength to the agent team.
>
> ---
> - _During training, “easy tasks” (those with a team accuracy ≥ 75%) are filtered out to focus on the leader’s coordination skills. This 75% threshold is heuristic and lacks theoretical grounding or systematic validation._
>
> **Response [Filtering Easy Takes]** The 75% threshold follows difficulty-filtering strategies used in prior works on GRPO-based training, e.g., [improving Data Efficiency for LLM Fine Tuning](https://arxiv.org/pdf/2506.05316), [DAPO](https://arxiv.org/pdf/2503.14476), [SKywork R1V2](https://arxiv.org/pdf/2504.16656). In our experiments, we observed that filtering out "easy" questions improved MLPO performance (Appendix B.3, Table 6). While some prior works use dynamic (online) filtering, we have a unique advantage in that agent-team solutions are computed before training. Thus, we can use these pre-computed responses to gauge the difficulty of each question (removing the need to compute difficulty during training). The most straightforward way is to set a hard threshold. In our early experiments we tested several thresholds and found that 75% was consistently the best performing threshold. As this filtering is informed by prior work, we view finding the "optimal" filtering value as out of the scope of our work. We merely wanted to outline that difficulty filtering helps in improving performance.

---

### Official Review · Reviewer_f3xb · 2025-11-01

**Soundness:** 3
**Presentation:** 3
**Contribution:** 2
**Rating:** 4
**Confidence:** 3

**Summary:**

This paper proposes a new concept called a multi-agent leader, then proposes a training method to tune the leader agent, aiming to improve the summarization and answer-seeking ability from multi-agent system rollouts. They test their proposed approach in three benchmarks and achieve a significant improvement compared to baselines.

**Strengths:**

- The paper introduces a novel and practical hierarchical multi-agent framework. Its main advantage is its computational efficiency, as it only requires training a single "leader" model while coordinating a team of fixed, untrained peer agents. This significantly reduces the training cost and complexity compared to approaches that require co-training multiple specialized models.
- The proposed Multi-agent guided Leader Policy Optimization (MLPO) is a novel contribution. It provides an effective method for training the leader to evaluate and synthesize diverse responses from the agent team without needing auxiliary value networks or explicit agent feedback, which simplifies the overall training pipeline.

**Weaknesses:**

- The benchmark (MMLU, BBH, MATH) does not seem to be related to a multi-agent system. They are knowledge-intensive tasks. I hope the author can provide some clarification on why they chose these benchmarks.
- Figure 2 seems not to be related to Figure 1, where in Figure 1, the data generation pipeline is online, i.e., the leader's feedback will return to the multi-agent system for new rollouts. If the leader's feedback cannot return to the multi-agent system, the leader will be degraded to a summarizer, which reduces the novelty of the task.
- The effectiveness of the framework appears to be sensitive to the choice of the leader model. The authors note in their analysis (Section 4.4) that while MLPO shows gains, the performance increase was less significant when using Gemma-2 and Llama-3.1 as leaders compared to Qwen-2.5. This suggests the method may require a highly capable base model (like Qwen-2.5) to act as an effective leader.

**Questions:**

N/A

---

> ### Author Response · Authors · 2025-11-21
> **Author Rebuttal**
>
> - _The benchmark (MMLU, BBH, MATH) does not seem to be related to a multi-agent system. They are knowledge-intensive tasks. I hope the author can provide some clarification on why they chose these benchmarks._
>
> **Response [Choice of benchmark]** Our work focuses on _multi-agent solutions_, not _multi-agent tasks_. This matches standard practice in the multi-LLM collaboration literature; prior works such as Multi-Agent Debate (Du et al., 2023), ACC-Collab (Estornell et al., 2024a), Reconcile (Chen et al., 2023), Mixture-of-Agents (Wang et al., 2024), REMAS, and others all evaluate multi-agent methods on general reasoning/factualness datasets such as BBH, MATH, and MMLU. The ultimate goal is to improve reasoning and factual accuracy, regardless of whether one adopts a multi-agent or single-agent solution (the previously cited works demonstrate that multi-agent solutions can yield higher efficacy on these domains, at the drawback of being more costly).
>
> ---
>
> - _Figure 2 seems not to be related to Figure 1, where in Figure 1, the data generation pipeline is online, i.e., the leader's feedback will return to the multi-agent system for new rollouts. If the leader's feedback cannot return to the multi-agent system, the leader will be degraded to a summarizer, which reduces the novelty of the task._
>
> **Response [Figures 1 and 2]**
> We would like to emphasise that indeed Figures 1 and 2 are meant to illustrate different settings. Figure 1 outlines interactions between the leader and the agent team at inference-time. Note that the leader's output is returned to agents after each round. Figure 2 outlines our training pipeline.
> To keep training practical, we precompute agent responses offline and train only the leader.
>
> A key focus of our work was on keeping training and deployment practical (e.g., reducing the number of models and generations required during training, or the ability to deploy our leader without the team at inference time).  Computing agent responses online would require holding multiple models in GPU memory and is substantially more expensive. However, since our pipeline is quite general, it would be easy to extend training to also include online agent responses when additional compute is available.
> In Section 4.3 we provided results for a modified version of MLPO in which the agent responses are based on the leader's feedback (called MLPO+). In Table 2, we see that including the leader feedback can result in a mild improvement (~1%-4% increase compared with MLPO). However, even without this additional feedback, MLPO attains a fairly significant improvement over all baselines. We thank the reviewer for their question and will clarify this distinction further in the final manuscript.
>
>
>
> **Response [Role of the Leader]** The leader is far more than a simple selector or summarizer. We provide several experiments to demonstrate this fact. Firstly, Figure 5 demonstrates that even when **all** agents are wrong, the leader can still produce the correct answer (effectively overriding the proposed solutions from the agent team), a behavior which would not be exhibited by a selector, summarizer, or aggregator. Secondly, we also compare to a so-called deferral leader, which attempts to select the best agent response from the team. Our trained leader significantly outperforms this type of leader; the leader also outperforms other baselines like SelectLLM, which is capable of selecting multiple responses from the agent team (see Table 1, page 6). In addition to these empirical results, the inference-time functionality of the leader is also different, as the leader also evaluates and analyzes the agent team's responses, providing explicit feedback for the next round.
>
> ---
> - _The effectiveness of the framework appears to be sensitive to the choice of the leader model. The authors note in their analysis (Section 4.4) that while MLPO shows gains, the performance increase was less significant when using Gemma-2 and Llama-3.1 as leaders compared to Qwen-2.5. This suggests the method may require a highly capable base model (like Qwen-2.5) to act as an effective leader._
>
> **Response [Sensitivity to Base Model]** We agree that the results are sensitive to the choice of base model (as discussed in Section 4.4). However, this type of sensitivity is expected for all fine-tuning-based approaches. We note that this also holds for the baseline methods such as GRPO, SCoRe, SelectLLM, ACC-Collab, and Self-Refine (Appendix Tables 7–8 for Gemma-2 and Llama-3 results), which all exhibit varying effectiveness for different models. Importantly we observe consistent improvements for all choices of base model and dataset (e.g., untrained leader vs the trained MLPO leader).

---

### Author Response · Authors · 2025-11-21
**General Response to Reviwers**

We want to thank you all for your thoughtful feedback! As a result, we have made a number of changes to the draft (marked in **blue** in the revised manuscript), including several new experiments and additional analysis. All new results are presented as tables in the draft to keep consistent with the format used here; we are happy to convert these results into plots if reviewers feel that would make them easier to interpret. We briefly highlight three of these new results here.

**Generalization across teams:**
We evaluate a leader trained with a heterogeneous team when paired at test time with homogeneous teams. We observe that the leader generalizes well across team compositions; in some cases, a leader trained on a heterogeneous team plus a homogeneous team at test time outperforms a leader trained directly on that homogeneous team, supporting our claim that exposure to diverse solutions at training time is beneficial. (Appendix B.2, Line 1178, Table 6.)

**Effect of team size:**
We train and evaluate a Qwen leader on teams of size 2, 3, and 4 (originally we presented results for size 3). We observe that generally larger teams yield better results, but the diverse team of 3 still consistently outperforms any homogeneous team of size 4.
We also evaluate our leader trained on 3 agents with teams of varying sizes (0 to 6 agents). Leader performance is generally robust to the number of agents, with slight degradation for larger team sizes (likely due to context overload or majority-drift), and in some weak-team settings it is best to deploy the leader solo (Appendix B.2, Line 1168, Table 5, Line 1222, Table 7).


**Inference-time Token-normalized comparisons**.
We compare accuracy vs. token budget (2.5k–20k tokens) across methods after two rounds of inference. For moderate to large token budgets (≥10k), MLPO with a team consistently outperforms all baselines, and the MLPO leader without a team consistently outperforms the GRPO model. However, GRPO or SCoRe are preferable only in low-budget settings (Appendix B.1, Line 956, Table 3).

---

### Meta-Review · Area_Chair_qGiZ · 2025-12-14

**Summary:**

The following concerns remain outstanding and, in my view, prevent the work from being sufficiently solid:
1. Limited evaluation scope.
2. Effectiveness when scaling to larger agent teams.
3. Generalization of the trained leader across different team compositions and tasks.
4. Trade-offs among performance, computational cost, and system complexity.

**Reviewer Concerns:**

**The following concerns have been addressed:**
1. The role and effectiveness of leader-to-team feedback.
2. The impact of filtering out easy training data.
3. Whether the observed performance gains primarily stem from supervised fine-tuning (SFT).
4. Clarity and completeness of the training details.

**The following concerns remain outstanding:**
1. Limited evaluation scope.
2. Effectiveness when scaling to larger agent teams.
3. Generalization of the trained leader across different team compositions and tasks.
4. Trade-offs among performance, computational cost, and system complexity.



**The following concern remains but is considered minor:**
1. Sensitivity to the choice of the leader model.
2. Theoretical foundation.

**Reviewer Scores:**

1. Reviewer f3xb is expected to maintain a score of 4, with a significant possibility of increasing it to 6.
2. Reviewer E79c is expected to maintain the score as 4.
3. Reviewer v9pM is expected to maintain the score as 4.
4. Reviewer Sy98 is expected to maintain the score as 4.
5. Reviewer D7xX is expected to maintain the score as 6.

---

### Decision · Program_Chairs · 2026-01-26

Reject